



# Characterization of primary and aged wood burning and coal combustion organic aerosols in environmental chamber and its implications for atmospheric aerosols

Amir Yazdani[1], Nikunj Dudani[1], Satoshi Takahama[1], Amelie Bertrand[2], André S. H. Prévôt[2], Imad El Haddad[2], and Ann M. Dillner[3]

[1]ENAC/IIE Swiss Federal Institute of Technology Lausanne (EPFL), 1015 Lausanne, Switzerland
[2]Laboratory of Atmospheric Chemistry, Paul Scherrer Institute, 5232 Villigen, Switzerland
[3]Air Quality Research Center, University of California Davis, Davis, California, USA

**Correspondence:** Satoshi Takahama (satoshi.takahama@epfl.ch) and Imad El Haddad (imad.el-haddad@psi.ch)

**Abstract.** Particulate matter (PM) affects visibility, climate, and public health. Organic matter (OM), a uniquely complex portion of PM, can make up more than half of total atmospheric fine PM. We investigated the effect of aging on secondary organic aerosol (SOA) concentration and composition for wood burning (WB) and coal combustion (CC) emissions, two major atmospheric OM sources, using mid-infrared (MIR) spectroscopy and aerosol mass spectrometry (AMS). For this purpose,

primary aerosols were injected into an environmental chamber and aged using hydroxyl (diurnal aging) and nitrate (nocturnal aging) radicals to reach an atmospherically-relevant oxidative age. A time-of-flight AMS instrument was used to measure high-time-resolution composition of non-refractory fine PM, while fine PM was collected on PTFE filters before and after aging for MIR analysis. AMS and MIR spectroscopy indicate an approximately three-fold enhancement of organic aerosol (OA) concentration after aging (not wall-loss corrected). The OM:OC ratios also agree closely between the two methods and

increase, on average, from 1.6, before aging, to 2, during the course of aging. MIR spectroscopy, which is able to differentiate among oxygenated groups, shows a distinct functional group composition for aged WB (high abundance of carboxylic acids) and CC OA (high abundance of non-acid carbonyls) and detects aromatics and polycyclic aromatic hydrocarbons (PAHs) in emissions of both sources. The MIR spectra of fresh WB and CC aerosols are reminiscent of their parent compounds with differences in specific oxygenated functional groups after aging, consistent with expected oxidation pathways for volatile

organic compounds (VOCs) of each emission source. The AMS mass spectra also show variations with source and aging that are consistent the MIR functional group (FG) analysis. Finally, comparison of the MIR spectra of chamber WB OA with that of ambient samples affected by residential wood burning and wildfires reveals similarities regarding the high abundance of organics, especially acids, and visible signatures of lignin and levoglucosan. This finding is beneficial to source identification of atmospheric aerosols and interpretation of their complex MIR spectra.

*Copyright statement.* TEXT





# 1 Introduction

Particulate matter (PM) affects visibility and climate (Hallquist et al., 2009). For examples, fine PM can play the role of cloud condensation nuclei (CCN) impacting cloud formation (McFiggans et al., 2004). PM can also considerably perturb transfer of different wavelengths of electromagnetic radiation by scattering or absorption phenomena (Seinfeld and Pandis, 2016). In addition, exposure to ambient fine PM is estimated to have caused 8.9 million premature deaths worldwide per year (in 2015; Burnett et al., 2018). Organic matter (OM), which constitutes up to 90 % of total fine atmospheric PM, is a key factor in aerosol-related phenomena (Russell, 2003; Shiraiwa et al., 2017). However, its chemical composition and formation mechanisms have not yet been fully characterized due to its compositional complexity (Kanakidou et al., 2005; Turpin et al., 2000).

Biomass burning particulate emissions (including those from residential wood burning, prescribed burning, and wildfire) are major contributors to total atmospheric OM with an ever-increasing importance due to rising wildfire activities (Westerling, 2016; DeCarlo et al., 2008; Sullivan et al., 2008). Biomass burning primary organic aerosols (POA) account for 16 % to 68 % of total OM mass in Europe (Puxbaum et al., 2007). Coal combustion (for electricity and heat generation) is another major POA source in China and some regions of Europe (Haque et al., 2019; Junninen et al., 2009), emitting considerable amounts of carcinogenic and mutagenic polycyclic aromatic hydrocarbons (PAHs) (Sauvain et al., 2003). Approximately 40 % world's electricity (and up to 66 % in China) is generated in coal-fueled power plants (World Coal Association). Biomass burning and coal combustion are also believed to be responsible for a large fraction of SOA, especially in winter when biogenic emissions are largely absent (Qi et al., 2019; Lanz et al., 2010; Zhang et al., 2020). Recent studies highlight the contribution of biomass burning by showing the predominance of carbon from non-fossil fuel origins in SOA even in industrial regions (Haddad et al., 2013; Beekmann et al., 2015). To date, primary biomass burning emissions have been investigated in several works (e.g. Johansson et al., 2004; Bäfver et al., 2011; Alves et al., 2011). However, SOA and their chemical composition have not been studied extensively until recently (e.g. Bertrand et al., 2017; Tiitta et al., 2016; Bruns et al., 2015) due to sophisticated experimental set-up requirements.

The determination of OA and SOA chemical composition, involves a large range of analytical and computational techniques. Aerosol mass spectrometry (AMS) and mid-infrared (MIR) spectroscopy are two methods capable of analyzing most of OA mass in addition to providing information about chemical class or the functional groups (FGs) (Hallquist et al., 2009). In an AMS, non-refractory aerosol is first vaporized (normally at 600 °C). Thereafter, the vaporized fraction is turned to ionized fragments and is then detected by the mass spectrometer to obtain on-line atomic composition of non-refractory aerosol. This method is well characterized for its capability to estimate elemental composition with high time resolution and detection limit (Canagaratna et al., 2007). There are, however, some known challenges: particle collection efficiency of aerodynamic lens section (Canagaratna et al., 2007); particle bounce back in the vaporizer (Kumar et al., 2018); potential reactions occurring in ion chamber (Faber et al., 2017); and, most importantly, extensive molecule fragmentation (Faber et al., 2017; Canagaratna et al., 2007) with the common ionization method (i.e. electron ionization, IE) that makes interpretation of AMS mass spectrum difficult.





MIR spectroscopy, which is commonly performed off-line on Polytetrafluorethylen (PTFE) filters, provides direct infor-
mation about FG abundances in OA collected on the filters. This information can be converted to OM, OC (organic carbon)
and the OM:OC ratio with the aid of statistical models and laboratory standards with few assumptions (Boris et al., 2019;
Ruthenburg et al., 2014; Reggente et al., 2016; Coury and Dillner, 2008). Recent studies show good agreement between MIR
OM and OC estimates and thermal optical reflectance (TOR) OC and residual OM method for monitoring networks (Boris
et al., 2019). The main advantage of MIR spectroscopy over other common methods is that it is relatively fast, inexpensive,
and non-destructive to the filter sample during analysis (Ruthenburg et al., 2014). However, sampling for MIR spectroscopy
is usually performed during at least several minutes resulting in lower temporal resolution (Faber et al., 2017). Moreover, the
presence of overlapping peaks complicates the interpretation of MIR spectrum. Collocated AMS and MIR measurements can
combine the advantages of both techniques and provide high-time-resolution measurements with FG quantification (e.g. Faber
et al., 2017; Chen et al., 2016; Frossard et al., 2014; Russell et al., 2009a; Chhabra et al., 2011b).

This study is one of the few examples of AMS and MIR spectroscopy being combined to provide a superior chemical
resolution for analyzing burning emissions in an environmental chamber and in the atmosphere. In this work, a series of
wood burning (WB) and coal combustion (CC) experiments were conducted in an environmental chamber at the Paul Scherrer
Institute (PSI). An AMS measured the chemical composition of OA throughout the aging process while fresh and aged fine
aerosols were collected on separate PTFE filters, making it possible to combine the measurements of the two methods. We
investigated the MIR spectra and FG composition of POA and SOA formed after diurnal and nocturnal aging processes in
relation to their parent compounds and the oxidation products of their identified VOCs. These results were combined with the
high-resolution AMS mass spectra to evaluate the consistency of the two techniques and to better understand differences in the
chemical composition caused by different emissions source and aging. Finally, the MIR spectra of the chamber biomass burning
samples were compared to that of some atmospheric burning-influenced aerosols collected at the Inter-agency Monitoring of
PROtected Visual Environments (IMPROVE) network (http://vista.cira.colostate.edu/improve/) to understand their similarities
and to develop a method for identification of atmospheric burning-influenced samples using MIR spectroscopy.

## 2  Methods

In the following sections, the experimental set-up (Sect. 2.1), on-line and off-line measurement techniques (Sects. 2.2 and
2.3), and atmospheric sample collection (Sect. 2.4) are described in detail. Thereafter, the statistical methods used for post-
processing are discussed in Sects. 2.5 and 2.6.

### 2.1  Laboratory experimental set-up and procedure

Four wood burning (WB) and six coal combustion (CC) experiments were conducted in a collapsible Teflon chamber of $6\,\mathrm{m^3}$ at
the Paul Scherrer Institute (PSI). We studied the effects of fuel source and diurnal/nocturnal aging on the chemical composition
of the emissions. The experimental set-up in this work was similar to that used by Bertrand et al. (2017) (Fig. S1).





For the WB experiments, three beech wood logs (approximately 2.5 kg) without bark, and additional 300 g of kindling were burnt in a modern wood stove (2010 model). The logs were ignited using three fire starters composed of wood shavings, paraffin and natural resins. The moisture content of the logs was measured to be around 11 %. Each burning experiment was started with a lighter followed by immediate closing of the burner door. Emissions past the ignition, in which kindling wood and starters were fully combusted, were injected into the chamber .

In the CC experiments, each time 300 g of bituminous coal from inner Mongolia (63 % carbon content) was burned. First, the ash drawer of the stove was loaded with kindling wood, which was ignited and served to ignite the coal. Thereafter, the wood was removed from the drawer after proper ignition of the coal. The emissions past the ignition phase were injected in the chamber via a single injection (similar to the procedure explained by Klein et al., 2018). In both CC and WB experiments, the injection was continued (from 5 to 25 minutes) until the measured concentration of primary organic aerosol by the high-
resolution time-of-flight (HR-TOF) AMS reached values of approximately 20 µg m$^{-3}$.

     WB and CC samples were extracted from the chimney and diluted using an ejector diluter (DI-1000, Dekati Ltd.) before injecting into the chamber. The lines from the chimney to the environmental chamber were heated to 413 K to limit semi-volatile compound condensation in the lines. The average temperature and relative humidity of the chamber after injection were maintained at 293 K and 55–60 %, respectively. The emissions were left static for 30 minutes in the chamber after
injection to ensure proper mixing and for sampling and characterizing the primary organic aerosol. Thereafter, emissions were aged using the hydroxyl or nitrate radical for simulating diurnal and nocturnal aging mechanisms.

     The OH radical was produced by photolysis of nitrous acid (HONO) continuously injected into the chamber, using UV lights (40 × 100 W, Cleo Performance, Philips). HONO was generated by reacting diluted sulfuric acid (H$_2$SO$_4$) and sodium nitrite (NaNO$_2$) in a gas flask and then was injected into the chamber after passing through a particle filter (the procedure is
explained by Taira and Kanda, 1990). Before aging, 1 µL of deuterated butanol-D9 (98%, Cambridge Isotope Laboratories) was injected into the chamber to measure the OH radical exposure (Barmet et al., 2012). The concentration of butanol-D9 was monitored by proton transfer reaction time-of-flight mass spectrometer (PTR-ToF-MS 8000, Ionicon Analytik). Emissions were aged for around four hours to reach OH exposures of $(2$–$3)\times10^7$ molec cm$^{-3}$ h corresponding to 20–30 hours of aging in the atmosphere (assuming a 24-hour average OH concentration of $1\times10^6$ molec cm$^{-3}$ in the atmosphere; Seinfeld and Pandis,
2016). For the nocturnal aging experiments, the NO$_3$ radical was produced by a single injection of O$_3$ and NO$_2$ in the chamber. The nitrate radical concentrations were estimated to be $(1.5$–$2.5)\times10^7$ molec cm$^{-3}$ for the first hour of aging.

     After each experiment, the chamber was cleaned by injecting O$_3$ for 1 hour in the chamber and irradiating with a set of UV lights while flushing with pure air (120 L min$^{-1}$ 737-250 series, AADCO Instruments Inc.). Then, the chamber was flushed with pure air in the dark for at least 12 hours (similar to the procedure described by Bruns et al., 2015).

## 2.2   On-line PM measurement

After the primary emission injections, PM emissions in the chamber were monitored using two on-line techniques. Non-refractory particle composition was measured at a temporal resolution of 30 seconds by a HR-TOF AMS (Aerodyne Research Inc.; DeCarlo et al., 2006) operating in V mode (mass resolution $\Delta m/m = 2000$), with a vaporizer temperature of 600 °C and



pressure of approximately $10^{-7}$ Torr, and EI operating at 70 eV, equipped with a 2.5 μm inlet aerodynamic lens. Data post
processing was performed in Igor Pro 6.3 (Wave Metrics) using SQuirrel 1.57 and Pika 1.15Z. The elemental and OM:OC
ratios were determined according to Aiken et al. (2008). The reported OA concentrations were not wall-loss corrected and a
unity collection efficiency was assumed.

A condensation particle counter (CPC, 3025A TSI) measured total particle number concentrations and a scanning mobility
particle sizer (SMPS, CPC 3022, TSI) measured particle size distribution. Particles were dried (Nafion, Perma Pure LLC)
upstream of the AMS, SMPS and CPC.

### 2.3  Off-line PM sampling and measurement

Primary and aged PM emissions were collected on separate PTFE filters (47 mm diameter Teflo® membrane, Pall Corporation)
for 20 minutes after injection of primary emissions into the chamber and after four hours of aging. The aerosol collection area
was limited to a circle with a diameter of 1 cm in the center of the filter using Teflon masking elements placed above and below
the filters (Russell et al., 2009b). Sampling on PTFE filters was performed at a flow rate of 8 L min$^{-1}$ using a flow system
composed of a sharp-cut-off cyclone (1 μm at nominal flow rate of 16 L min$^{-1}$) and a silica gel denuder. Hereafter, these
PTFE filters are referred to by their fuel, and oxidant: e.g. WB_OH refers to the filters corresponding to the WB experiments
aged with OH. After sampling, filters were immediately stored in filter petri dishes at 253 K before MIR analysis to minimize
volatilization and chemical reactions. The PTFE filters were analyzed using a Bruker-Vertex 80 FT-IR instrument equipped
with a $\alpha$ deuterated lanthanum alanine doped triglycine sulfate (DLaTGS) detector, at a resolution of 4 cm$^{-1}$.

### 2.4  Atmospheric samples (IMPROVE network)

Particulate matter with diameter less than 2.5 μm (PM$_{2.5}$) was collected on PTFE filters (25 mm diameter Teflo® membrane,
Pall Corporation) every third day for 24 hours, midnight to midnight, at nominal flow rate of 22.8 L min$^{-1}$ during 2011 and
2013 at selected sites in the IMPROVE network (approximately 3050 samples from 1 urban and 6 rural sites in 2011 and
4 urban and 12 rural sites in 2013). The PTFE filters were analyzed using a Bruker-Tensor 27 FT-IR instrument equipped
with a liquid nitrogen-cooled, wide-band mercury-cadmium-telluride (MCT) detector, at a resolution of 4 cm$^{-1}$. In this work,
atmospheric samples were divided into four mutually exclusive sub-groups: urban, rural, residential wood burning, and wildfire
(residential wood burning, and wildfire samples were identified by Bürki et al., 2020). Bürki et al. (2020) separated and
identified the burning-influenced samples in the same dataset using cluster analysis and subsequent analysis of the clusters.
They divided the burning-influenced samples into residential wood burning and wildfire sub-categories by extending further
down the hierarchical tree. The residential wood burning sub-category was labeled according to its occurrence during winter
months at Phoenix, AZ (Bürki et al., 2020), where residential wood burning commonly takes place (Pope et al., 2017). The
wildfire sub-category was labeled due to its occurrence during a known fire event (Rim Fire, CA, 2013).





### 2.5 Post-processing of MIR spectra to identify and quantify functional groups in laboratory and IMPROVE samples

After obtaining the MIR spectra of the laboratory and ambient samples, they were processed the same. The post-processing enabled quantification of alcohol COH (aCOH), carboxylic acid (COOH), aliphatic CH (aCH), and non-acid carbonyl (naCO) functional groups and identification of PAHs, organonitrates, levoglucosan, inorganic sulfate and nitrate. In this section, the methods used for spectral post-processing are described in detail.

#### 2.5.1 Baseline correction

Baseline correction was performed to eliminate contribution of background drift, light scattering by filter membrane and particles, and absorption by carbonaceous material due to electronic transitions (Russo et al., 2014; Parks et al., 2019). For this purpose, we used a smoothing spline method similar to the approach taken by Kuzmiakova et al. (2016). In this method, a cubic smoothing spline was fitted to the raw spectra (excluding organic FG bands) and then was subtracted from the them to obtain the net absorption due to FG vibrations at each wavelength. The current version extends the baseline correction algorithm by

Kuzmiakova et al. (2016) (limited to $1500\text{--}4000~\mathrm{cm}^{-1}$) to $400\text{--}4000~\mathrm{cm}^{-1}$ range.

#### 2.5.2 Blank subtraction

Although Teflon filters are optically thin in the MIR range, they have several absorbing bands due to the $\mathrm{C-F}$ bond vibrational modes (e.g. at $1000\text{--}1320~\mathrm{cm}^{-1}$; Quarti et al., 2013). These bands overlap with some organic and inorganic function group bands and limit the information that can be extracted from the analysis of OA on PTFE filters. To mitigate this issue, a scaled

version the of a baseline-corrected blank filter spectrum (the contribution of filter membrane scattering was excluded and only PTFE absorptions were maintained) was subtracted from the baseline-corrected sample spectra to retrieve some of the overlapping features (Fig. S2). The scaling procedure compensates for the variation in blank filter absorbances due to factors, such as non-uniformity in PTFE membrane morphology due to manufacturing variability and/or aerosol loading differences between filters and within the same filter (Debus et al., 2019; Quarti et al., 2013) by scaling the $\mathrm{C-F}$ peak at $1210~\mathrm{cm}^{-1}$. This

approach is different from those taken by Takahama et al. (2013) and Maria et al. (2003) that subtracted a scaled raw spectrum of a blank filter from sample spectra before baseline correction. The blank subtraction algorithm allowed us to identify bands related to aromatics and PAHs at $690\text{--}900~\mathrm{cm}^{-1}$ (Centrone et al., 2005), organonitrates ($\mathrm{RONO_2}$) at $850~\mathrm{cm}^{-1}$, alcohol CO stretching at $1050~\mathrm{cm}^{-1}$, levoglucosan bands at $860\text{--}1050~\mathrm{cm}^{-1}$, inorganic sulfate and nitrate bands at $620$ and $1400~\mathrm{cm}^{-1}$, and also allowed us to better quantify the carbonyl absorbances at around $1700~\mathrm{cm}^{-1}$.

#### 2.5.3 Quantifying organic functional groups

After baseline correction and blank subtraction, the multiple peak-fitting algorithm described by Takahama et al. (2013) and implemented by Reggente et al. (2019b), functioning based on non-linear least squares analysis, was applied to the spectra to obtain major FG abundances of aCOH, COOH, aCH, and naCO (Yazdani et al., 2020, manuscript in preparation). The $\mathrm{RONO_2}$ group abundances were not quantified due to extensive overlap of its absorbances with other compounds and in order to keep





the MIR estimates consistent with those of AMS, for which only total (organic plus inorganic) nitrate was estimated. After estimating FG abundances, the O:C, H:C and OM:OC ratios were calculated with few assumptions about the number of carbon atoms attached to each FG (refer to Chhabra et al., 2011b; Russell, 2003; Maria et al., 2002). A few other peaks attributed to aromatics and PAHs were analyzed qualitatively due to lack of calibration models. In addition to the common FG analysis, we used MIR fingerprint features to identify relevant substances to biomass burning (e.g. levoglucosan and lignin).

### 2.6   Dimensionality reduction of AMS mass spectra

While AMS provides a well-characterized, time-resolved measurement of OA aging, extensive fragmentation of molecules, high number of ion fragments, and collinearity of ion fragment intensities make the interpretation of AMS mass spectra complex (Faber et al., 2017; Canagaratna et al., 2007). We used principal component analysis (PCA; Hotelling, 1933) to reduce the dimensionality of the AMS mass spectra in order to identify the most important drivers of variability in the spectral data
and their connection with FG composition of OA. The advantage of PCA analysis over analysis of the normalized conventional mass fragments (e.g. $f_{43}$ and $f_{44}$ in Fig. 5) is that PCA loads the highly correlated fragments ions onto the same principal components (PCs) that are orthogonally oriented to each other, thereby reducing redundancy among dimensions. Furthermore, PCA describes the range of variability spanned specifically by this data set, accentuating smaller variations that might be lost using the conventional ranges spanned by normalized mass fragment analysis.

Before applying PCA, mass spectra at each measurement were normalized using the corresponding OA concentrations to eliminate signal variability due to change in OA concentration. Correlation matrix of the normalized AMS mass spectra (correlation of ion intensities at different m/z values) shows that signals at several m/z values are correlated (Fig. S3). Thus, the dimensionality of the mass spectra can be reduced considerably without significant loss of information. Our data matrix, $\mathbf{X}$, is a $i \times j$ matrix with $i$ observations (2979 AMS measurements) and $j$ variables (335 fragment ions) with a rank $l$ ($l \leq \min\{i, j\}$).
The columns of $\mathbf{X}$ is centered to avoid intercepts in scores. PCA calculation was performed by singular value decomposition of the centered data matrix (Abdi and Williams, 2010), $\mathbf{X} = \mathbf{P}\boldsymbol{\Delta}\mathbf{Q}^T$, where $\mathbf{P}$ is the $i \times j$ matrix of left singular vectors, $\mathbf{Q}$ is the $j \times l$ matrix of right singular vectors, known as loadings, and $\boldsymbol{\Delta}$ is the diagonal matrix of singular values ($\boldsymbol{\Delta}^2$ is the diagonal matrix of eigenvalues of covariance matrix, $\mathbf{X}^T\mathbf{X}$). The $i \times l$ matrix of factor scores, $\mathbf{F}$, is obtained as $\mathbf{F} = \mathbf{P}\boldsymbol{\Delta}$.

### 3   Results and discussions

In this section, peak fitting is applied to MIR spectra to estimate major FG abundances and to obtain detailed information about oxygenated species formed with aging and differences in FG composition between OA of different sources (Sects. 3.1 and 3.3). In addition, the MIR spectra of primary and aged OA emissions are compared to that of the fuel sources for identifying the chemical similarities between POA and fuel sources and for better understanding the important oxidation pathways for SOA formation (Sects. 3.2 and 3.4). In Sec. 3.5, AMS mass spectra are analyzed using PCA to understand the factors deriving the
spectral variability and how they relate to FG composition. Finally, the MIR spectra of chamber WB aerosols are compared



to that of previously identified atmospheric burning-influenced samples and a new method for identifying biomass burning aerosols using MIR spectroscopy is proposed (Sect. 3.6).

### 3.1 Wood burning – functional group composition

The primary WB aerosols have high abundance of the aCH group (Figs. 1a and S5a). This FG, which absorbs at 2800–3000
cm$^{-1}$ region in MIR spectrum (Fig. 2a), constitutes around 50 % of the total mass of fresh WB OA (Fig. 1a). The aCOH group, which appears as a broad peak around 3400 cm$^{-1}$ (Fig. 2a), has the second highest concentration (around 30 % of the total mass; Figs. 1a). This FG is also ubiquitous in hemicellulose, cellulose, and lignin – three main components that constitute 20–40, 40–60, and 10–25 wt. % of lignocellulosic biomass, respectively (McKendry, 2002). The COOH group, which appears as a broad peak in 2400–3400 cm$^{-1}$ and a sharp carbonyl peal at approximately 1700 cm$^{-1}$ (Fig. 2a), is the third abundant FG in fresh WB aerosols and constitutes 10–20 % of the OA mass (Figs. 1 and 2a). The primary WB samples have OM:OC ratios ranging from 1.6 to 1.8 (Figs. 2b and S5b).

AMS and MIR estimates of OA concentration are highly correlated ($r^2 = 0.92$; refer to Yazdani et al., 2020, manuscript in preparation, for detailed comparison) and show an almost three-fold increase in OA mass concentration with aging. The relative aCH abundance decreases substantially (by up to 30 %) in WB aerosols with aging (Figs. 1a and S5a). The relative decrease of the aCH abundance is less prominent when aerosols are aged with the nitrate radical, probably due to the fact that organonitrates are excluded from quantitative analysis thus the total OA concentration is underestimated and also because only a limited number of precursors react with the nitrate radical. The aCH profile changes from the superposition of sharp peaks (observed for long-chain hydrocarbons) in primary WB spectra to broad peaks (observed in the spectra of oxygenated species) in aged WB spectra (Fig. 2a; Yazdani et al., 2020). The aCOH relative abundance in WB aerosols also decreases with aging (Fig. 1a). Relative and absolute abundances of COOH increase significantly in WB aerosol with aging, suggesting carboxylic acid formation to be the dominant VOC oxidation pathway for biomass burning (Fig. 1). The aged WB samples with high carboxylic acid concentration have a broad OH peak ranging from 2400 to 3400 cm$^{-1}$ and their carbonyl absorption frequency is on the lower end of its range (approximately 1708 cm$^{-1}$ compared to 1715 cm$^{-1}$ for ketone carbonyl; Fig. 2a) due to weakening of C=O bond in dimerized acids (Pavia et al., 2008). Phenol, methoxyphenols, and naphthalene are among the most important SOA precursors, based on their SOA yields, present in wood smoke reported by Bruns et al. (2016) and Stefenelli et al. (2019). The high abundance of COOH in the aged WB OA of this study is consistent with the considerable carboxylic acid formation reported from these precursors (Chhabra et al., 2011a; Kautzman et al., 2010; George et al., 2015). The aged WB samples have the highest OM:OC ranging from 1.8 to 2.1 (Figs. 2b and S5b), with high concentrations of COOH.

MIR spectroscopy is able to distinguish between organic and inorganic nitrate due to differences in their absorption frequencies (Day et al., 2010). In this work, we have investigated the variations of the RONO$_2$ bands qualitatively by analyzing its absorbances at 1630, 1273, and 850 cm$^{-1}$ (Day et al., 2010). These peaks are negligible in the primary WB aerosols. However, their absorbances (thus the abundance of organonitrates) increase in WB aerosols aged by both hydroxyl and nitrate radicals. with much more prominent contributions, when the nitrate radical is used (Fig. 2a). This suggests that RONO$_2$ formation could be an important SOA formation pathway from WB emissions, which, nevertheless, we do not account for quantitatively. Weak





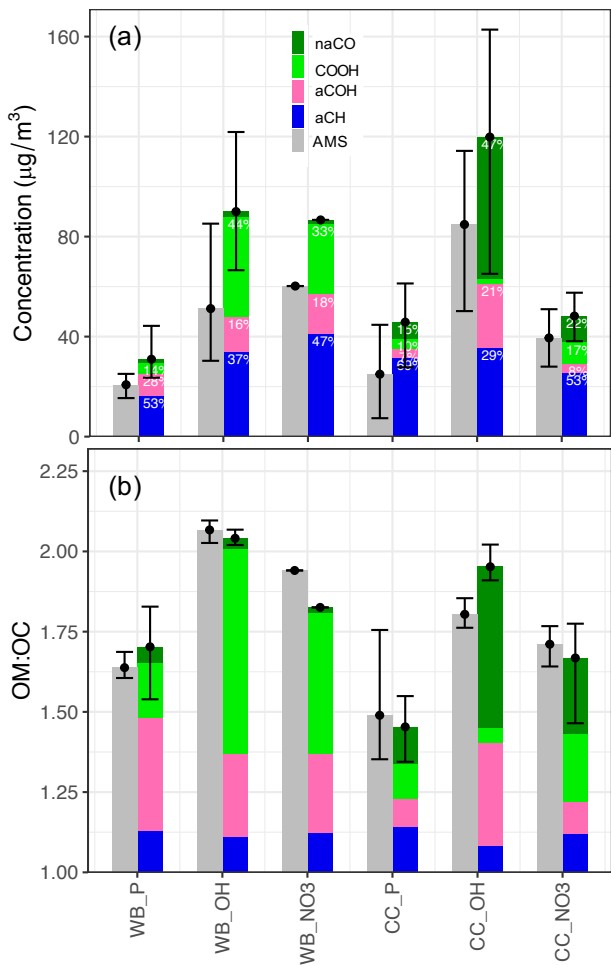

**Figure 1.** (a) Bar plot of averaged MIR (separated by functional group) and AMS OA concentration estimates without wall loss correction. The contribution of each functional group (with contribution $> 5\%$) to total OA, the type of aerosol (P: primary), emission source (WB and CC), and oxidant used for aging (OH: hydroxyl radical, NO$_3$: nitrate radical) are indicated for each category. (b) Bar plot of averaged MIR and AMS OM:OC estimates separated by contribution of each functional group. The error bars show the maximum and minimum concentration and OM:OC values for each category. For the estimates of each individual experiment refer to Fig. S5.

signatures of aromatics or PAHs are visible at 690–900 cm$^{-1}$ in the MIR spectra of both primary and aged WB aerosols. The absorption in this region appears as a single peak centered around 754 cm$^{-1}$. The intensity of this peak is correlated ($r^2 = 0.70$) to the concentration of fragments in the AMS mass spectra that were reported to be attributed to aromatics (Bruns et al., 2015; Pavia et al., 2008) (Fig. S8). AMS results suggest aromatics and PAHs in WB aerosols constitute up to 8 % of total OA. However, underestimation is possible due to incomplete list of ion fragments considered in this work and fragmentation

of oxygenated aromatics and PAHs during EI ionization (Supplement Sect. S5).





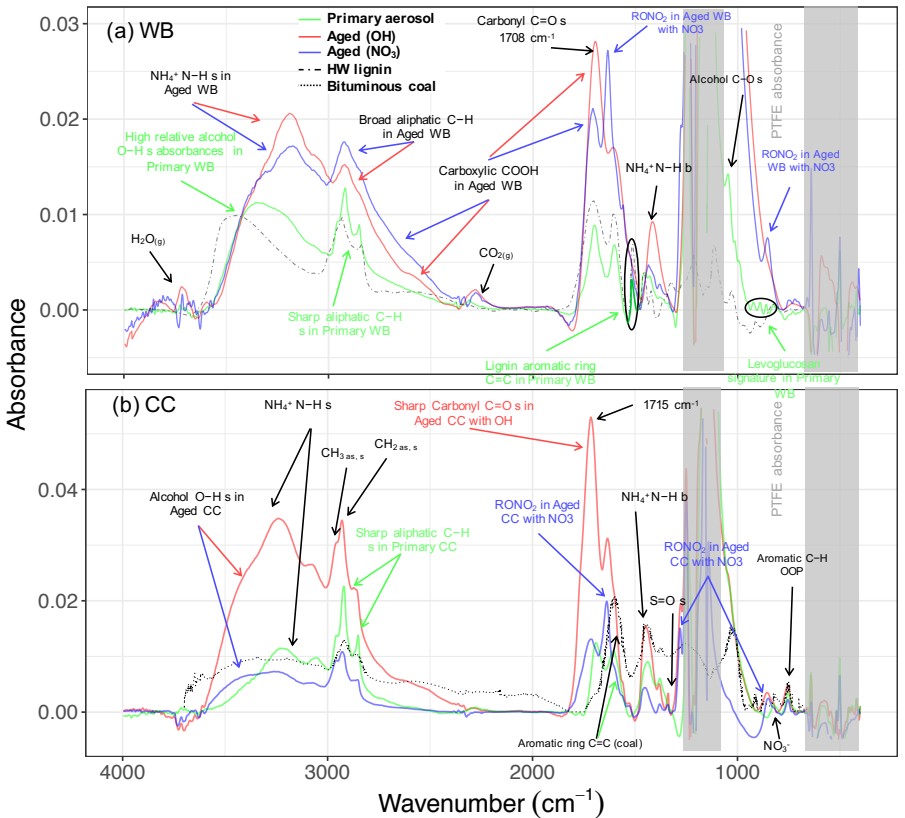

**Figure 2.** The MIR spectra of primary and aged (with OH and NO$_3$) wood burning (WB) and coal combustion (CC) OA and their parent compounds: lignin and bituminous coal. The dash-dotted line demonstrates the spectrum of solvent-extracted hardwood (HW) lignin (in KBr pellet) taken from Boeriu et al. (2004). The dotted line shows the spectrum of pulverized bituminous coal taken from Zhang et al. (2015). Stretching and bending vibrations are denoted by "s" and "b", respectively and important absorbances for each category are color-coded.

## 3.2 Wood burning – primary signatures

Using the MIR signatures of levoglucosan in primary WB aerosols (Sect. S6, and Fig. S9), it is estimated that 22–48 % of the aCOH absorbance and 15–29 % of total OA mass is due to the presence of levoglucosan as a main product of high temperature pyrolysis of cellulose and an important biomass burning marker (Shen and Gu, 2009; Puxbaum et al., 2007; Hennigan et al., 2010). The reported range is consistent with measurements by a thermal desorption aerosol gas chromatograph coupled to a HR-TOF AMS (TAG-AMS) TAG-AMS obtained by Bertrand et al. (2018a). The abundance of the AMS m/z 60 fragment ion, related to Levoglucosan fragmentation, (Schneider et al., 2006) and its MIR absorbances (used for the first time in this work) are highly correlated ($r^2 = 0.76$; Fig. S10), both showing a consistent decrease with aging regardless of the type of oxidant (Fig S10). The contribution of levoglucosan to total OA mass is estimated to decrease to less than 5 % after aging. This is





consistent with the significant enhancement in OA mass with SOA formation and the degradation of Levoglucosan through its

reaction with OH and its loss to the chamber walls Bertrand et al. (2018b); Hennigan et al. (2010); Zhao et al. (2014).

The MIR spectra of hard wood lignin and fresh WB aerosols are very similar, suggesting the presence of similar molecular

structures between primary WB aerosol composition and its parent compound, lignin (Fig. 2a). One specific aspect of this

similarity is the high absorbances of the aCOH group in both samples. By comparing the MIR spectra of fresh WB aerosols

with that of lignin (Supplement Sect. S7), it can be inferred that the sharp peak at 1515 $cm^{-1}$ and a part of the broader peak

at 1600 $cm^{-1}$ are attributed to aromatic rings in lignin structure (Hergert, 1960; Yang et al., 2007; Derkacheva and Sukhov,

2008) observed also in the spectrum of lignin monomers such as coniferyl alcohol (Bock and Gierlinger, 2019) and pyrolysis

products of lignin with the same substitution pattern (Duarte et al., 2007; Simoneit et al., 1993). The peak at 1515 $cm^{-1}$,

however, diminishes with aging, implying a change in aromatic ring substitution or ring opening by oxidant attack. The peak

around 1600 $cm^{-1}$ is suppressed by the $RONO_2$ peak around 1630 $cm^{-1}$ after aging. Overall, our analysis shows that the large

majority of the primary OA from WB emissions is composed of anhydrous sugars and lignin pyrolysis products.

### 3.3 Coal combustion – functional group composition

Peak fitting analysis suggests that the aCH group from all compounds containing this FG (alkanes and other compounds)

constitutes around 60–80 wt. % of primary CC OA – a larger fraction compared to fresh WB OA (Figs. 1b and S5b). Abundance

of short and long-chain alkanes has also been reported in VOC emissions of bituminous coal (Liu et al., 2017; Klein et al.,

2018). The naCO group is the second most abundant FG in the fresh CC OA, constituting, on average, 15 % of its mass (Fig.

1a), much higher than their contribution to WB emissions. This functional group has been reported to constitute 5–15 % wt.

of bituminous coal VOC emissions (Liu et al., 2017; Klein et al., 2018). The concentration of nd COOH groups is usually low

in the primary CC aerosols (around 10 %; Fig. 1a). The primary CC aerosols are estimated to have the lowest OM:OC ratios

(1.35–1.5) among all samples, which is justified by their strong aCH absorbances (Fig. 2b). Inorganics (ammonium, sulfate,

and nitrate) have prominent absorbances in the MIR spectra of fresh (and aged) CC aerosols, which have not been observed in

the case of WB aerosols. The high abundance of inorganic salts can be attributed to the sulfur and nitrogen that are present in

bituminous coal (Vasireddy et al., 2011).

In the aged CC aerosols, the relative abundance of the aCH group decreases drastically (on average 40 %), especially when

the hydroxyl radical is used as oxidant, despite an almost three-fold increase in the OA mass concentration (without wall-

loss correction; Figs. 1a and S5a). On the other hand, the abundance of the naCO carbonyl group increases by up to 40 %,

suggesting that carbonyl production is the dominant oxidation pathway for VOCs of CC emissions, predominantly when the

hydroxyl radical is used (Fig. 1a). This observation is consistent with high carbonyl abundance in SOA formed from the OH

oxidation of alkane precursors, which are abundant in CC SOA, at close OH exposures to those in this work (Lambe et al., 2012;

Lim and Ziemann, 2009). This does not, however, preclude contribution of oxidation products of aromatic compounds, which

are also abundant in CC VOCs (Liu et al., 2017; Klein et al., 2018), in aged CC OA. The absence of the broad carboxylic acid

OH peak for these samples, and the peak location of the carbonyl group (1715 $cm^{-1}$) in their MIR spectra (Fig. 2b) suggest

that the majority of the carbonyl group in the OH-aged CC aerosols is ketone (Pavia et al., 2008). An increase in the aCOH





abundance is also observed in the CC aerosols with aging. This increase is more remarkable than that in the WB aerosols (Fig.
295 1).

The RONO$_2$ signature becomes clearly visible in aged CC aerosols, when the nitrate radial is used as oxidant (Fig. 2b)
or when the hydroxyl radical is used in the presence of NO$_x$ (> 50 ppb). This observation is consistent with VOC oxidation
pathways proposed by Kroll and Seinfeld (2008) and measured by Ayres et al. (2015). In addition, a new peak at 1350 cm$^{-1}$
that can be attributed to the S=O group in sulfonates (Pavia et al., 2008) appears in some aged CC aerosols (Fig. 2b).

The aromatic CH OOP absorption appears as a relatively stronger peak at 754 cm$^{-1}$ compared to WB aerosols in the spectra
of CC aerosols and bituminous coal (Fig. 2b; Sobkowiak and Painter, 1992; Sobkowiak and Painter, 1995). The normalized CH
OOP absorbances by total OA mass are on average higher for CC aerosols compared WB, suggesting a higher contribution this
FG. The concentration of AMS fragment ions corresponding to PAHs suggest that aromatics and PAHs of CC aerosols account
for up to 7 % of total OA mass although some underestimation is possible due to limited number of fragment ions considered
and fragmentation of oxygenated aromatics and PAHs upon ionization (Supplement Sect. S5). These measurements suggest
lower concentrations of aromatic compounds in CC OA compared to CC VOC emissions measured by Liu et al. (2017) and
Klein et al. (2018).

The aged CC aerosols have slightly lower OM:OC ratios compared to the aged WB aerosols (Fig. 1b). For both emission
sources, aerosols aged with the hydroxyl radial have higher OM:OC ratios (approximately 0.2) than those aged with the
nitrate radical (Fig. 1b). This can be attributed the different rate constants of VOC reactions with nitrate and hydroxyl radicals
(Ziemann and Atkinson, 2012). Furthermore, organonitrates, which are abundant when aerosols are aged with the nitrate
radical, are not considered either in AMS OA estimates or in MIR peak fitting. This exclusion causes under-prediction of both
the OM:OC ratio and OA concentration estimates when the nitrate radical is used.

### 3.4  Coal combustion – primary signatures

Bituminous coal contains 69–86 wt. % carbon and its chemical structure is formed by highly substituted rings that are connected
by alkyl or ether bridges (oxygen or sulfur) (Vasireddy et al., 2011). The MIR spectra of primary CC aerosols have weaker
aromatic C=C absorbances at 1610 cm$^{-1}$ than bituminous coal (Fig. 2b). This peak is suppressed by the RONO$_2$ peak at 1630
cm$^{-1}$ with aging (Fig. 2b). The carbonyl peak (1710 cm$^{-1}$) observed in the spectra of the primary CC aerosols is absent in
that of bituminous coal (Fig. 2b), suggesting carbonyl formation during coal combustion. Zhang et al. (2015) also reported
a similar carbonyl generation for coal oxidation in high temperatures. We also observe that the aCH peaks are considerably
sharper in the spectra of fresh CC aerosols than the pulverized coal (Fig. 2b), suggesting a higher relative contribution of this
FG in primary CC OA than in bituminous coal.

The analyses of Sects. 3.1–3.4 suggest that aerosol source (WB and CC) and type (primary and aged with different oxidants)
are both drivers of variability in the MIR spectra. Furthermore, we found that aged aerosols of WB and CC, have distinguishable
functional group compositions, suggesting that MIR spectra retain source class information at least up to the studied levels of
aging.





## 3.5 Comparison between AMS mass spectra and FG composition of OA

In this section, we investigate the main differences in bulk chemical composition of the primary and aged emissions measured by the AMS and their FG content. We applied PCA to the AMS mass spectra to facilitate their interpretation and to better

understand the major sources of variability in the spectra and their connection with FG composition of OA. Moreover, we projected AMS PMF (positive matrix factorization) factors obtained by Elser et al. (2016) and Aiken et al. (2009) (High Resolution AMS Spectral Database: http://cires.colorado.edu/jimenez-group/HRAMSsd/) onto the PC space to compare and contrast the AMS spectra obtained from the chamber experiments with AMS PMF factors of atmospheric samples. By applying PCA, dimensionality of the normalized AMS mass spectra was reduced from 330 fragment ions to 3 PCs, while explaining

around 91 % of variance in the mass spectra (Table 1).

**Table 1.** Importance of the first three principal components.

|  | PC1 | PC2 | PC3 |
|---|---|---|---|
| **Standard Deviation** | 0.046 | 0.019 | 0.01461 |
| **Proportion of Variance** | 0.715 | 0.120 | 0.071 |
| **Cumulative Proportion** | 0.715 | 0.835 | 0.907 |

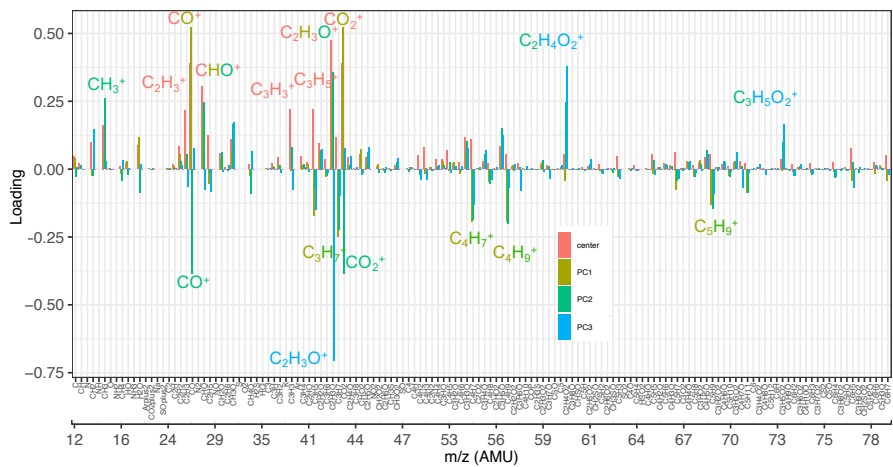

**Figure 3.** Loadings of the first three principal components and the normalized mean AMS mass spectrum (shown up to m/z 80). Fragment ions with high positive/negative loadings are indicated by their formula and are color-coded for each PC.

As can be seen from Fig. 3, the mean (center) mass spectrum, which represents the average of all mass spectra, has its highest values at a few ion fragments: $C_2H_3O^+$, $CO^+$, $CO_2^+$, $CHO^+$, $C_2H_3^+$, $C_3H_3^+$, and $C_3H_5^+$. The loadings, which are informative about the most important variations in the mass spectra, have also high values at a few mass fragments, making their interpretation simple. The first PC, which explains around 72 % of variance alone, has high positive loadings of $CO_2^+$ and





$CO^+$, $CHO^+$, and $C_2H_3O^+$ fragment ions (Fig. 3). The $CO_2^+$ fragment has been previously ascribed to mono and di-carboxylic acids (Canagaratna et al., 2007; Faber et al., 2017; Russell et al., 2009a; Frossard et al., 2014). The $CO^+$ fragment is estimated directly from $CO_2^+$ (Aiken et al., 2008) and the other mentioned ion fragments represent other oxygenated species (Faber et al., 2017; Chhabra et al., 2011a). The high loadings of major oxygenated ion fragments and the empirical observation that primary and aged samples are separated along PC1 axis (Fig. 4) suggest that the first PC indicates the general direction of aging and

the extent of oxidation, which appear to be the major sources variation in the data (based on high explained variance by PC1). As can be seen from Fig. 4, PC1 scores are the lowest for primary CC and WB aerosols, respectively and increase with aging for both WB and CC samples. The higher oxidation state of aged WB aerosols compared to aged CC (i.e., higher PC1 score) aerosols, is consistent with the high abundance of COOH and naCO FGs in the aged WB and CC aerosols, respectively. The WB and CC samples that are aged with the hydroxyl radical have higher PC1 scores than those aged of the nitrate radical,

indicating the former are more oxidized. The order of PC1 scores across the samples is reminiscent of the order of OM:OC ratios discussed earlier.

The second PC, which explains about 12 % of variance, mainly contrasts primary WB (with high abundance of levoglucosan and lignin) with primary CC (with high abundance of of aCH) aerosols by high positive loadings of $C_2H_4O_2^+$, representing levoglucosan fragmentation (Schneider et al., 2006), and $C_8H_9O_2^+$ (not in the range of Fig. 3), representing lignin fragmentation

(Tolbert and Ragauskas, 2017; Saito et al., 2005), and negative loadings of $C_xH_y$ fragments, attributed to the aCH group (Faber et al., 2017). In addition, this PC has high positive loadings of $C_2H_3O^+$, $CHO^+$ mass fragments and high negative loadings $CO^+$ and $CO_2^+$ (Fig. 3), differentiating between the COOH group and other oxygenated FGs such as aCOH (Faber et al., 2017). The primary CC aerosols have the lowest PC2 scores due to high aCH content. Their PC2 scores increase with aging (Fig. 4a), indicating production of $CHO^+$ and $C_2H_3O^+$ fragments (non-acid oxygenated FGs, such as the naCO and the aCOH

groups) and decrease slightly before the end of aging probably due to production of $CO_2^+$ (the COOH group) outweighing other oxygenated fragments. On the other hand, the primary WB aerosols have the highest PC2 scores (Fig. 4a) due to high abundance aCOH, levoglucosan, and lignin. The PC2 scores reduce drastically with aging for WB aerosols, especially when the hydroxyl radical is used, suggesting formation of the COOH group and degradation of levoglucosan and lignin (Fig. 4a). PC2 scores are higher for WB and CC samples aged with the nitrate radical compared to those aged with the hydroxyl radical

(Fig. 4a) due to lower concentration of the $CO_2^+$ fragment ion (Figs. 5 and S6).

The first two principal components, which together explain 84 % of total variance, are able to separate aerosols according to their oxidation state (PC1) and their source (PC2) and show that the trajectories of CC and WB OA start to converge, especially when with aging the hydroxyl radical (Fig. 4a). This observation implies that the spectral differences between samples of the same or different category decreases with aging despite the higher oxidation state (contribution to PC1) of the aged WB aerosols

compared to the aged CC. The primary WB aerosols on PC1–PC2 biplot (Fig. 4a) are located close to biomass burning OA (BBOA) factors obtained by Aiken et al. (2009) and Elser et al. (2016), highlighting their chemical similarity. The same is observed for CC aerosols with coal combustion OA (CCOA) and hydrocarbon-like OA (HOA) factors. The OH-aged aerosols are, however, more similar to the semi-volatile oxygenated OA (SV-OOA) factor, representing less oxygenated OA (Fig. 4a) and have considerably lower PC1 compared to the aged OA (OOA) factor by Aiken et al. (2009), which represents aged SOA





(i.e., more aged than SV-OOA; not in the range of Fig. 4). The location of the samples in $f_{44}$–$f_{43}$ plot (far from the triangle vertex; Fig. 5) also suggests a moderate extent of oxidation for the aged WB and CC samples.

The third PC, which explains 7 % of variance, mainly separates the aged aerosols based on the type of oxidant used and has high negative loading of $C_2H_3O^+$ and positive loadings of the $C_2H_4O_2^+$ and $C_3H_5O_2^+$ fragments (Fig. 3). PC3 scores have the highest values for the primary WB aerosols due to abundance of levoglucosan and decrease with aging (Fig. 4b) due to

degradation of levoglucosan and generation of the $C_2H_3O^+$ fragment (Fig. 5). PC3 scores are lower for the primary CC aerosols due to absence of levoglucosan and decrease further with aging due generation of the $C_2H_3O^+$ fragment. For both sources, the aerosols aged with the nitrate radical have considerably lower PC3 scores (Fig. 4b) due to high relative abundance of the $C_2H_3O^+$ fragment (Figs. 5 and S6). The aged CC aerosols with high abundance of naCO also have relatively low PC3 scores (Fig. S7b), suggesting higher $C_2H_3O^+$ concentration for those samples (also observed from Fig. 5). This observations suggests

the species formed during aging with the nitrate radical and the naCO group produce higher concentrations of the $C_2H_3O^+$ fragment.

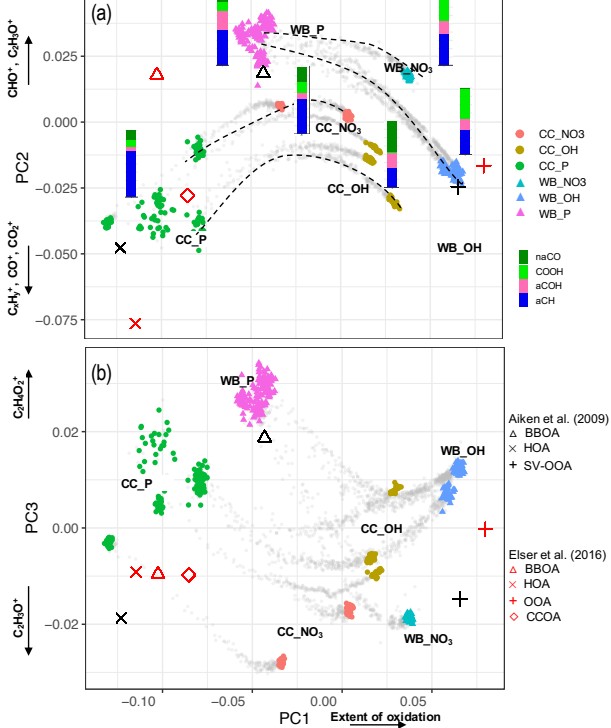

**Figure 4.** Biplots of PC2–PC1 (a), and PC3–PC1 (b) scores. The AMS measurements corresponding to filter sampling periods are color-coded by category. The AMS measurements out of filter sampling periods are illustrated by gray dots and some oxidation trajectories are indicated by dashed curves. AMS BBOA, CCOA, HOA, and SV-OOA factors (Elser et al., 2016; Aiken et al., 2009; Ulbrich et al., 2009) are projected onto PCs for comparison. Average FG composition for each category estimated from the MIR spectra is shown beside the category.





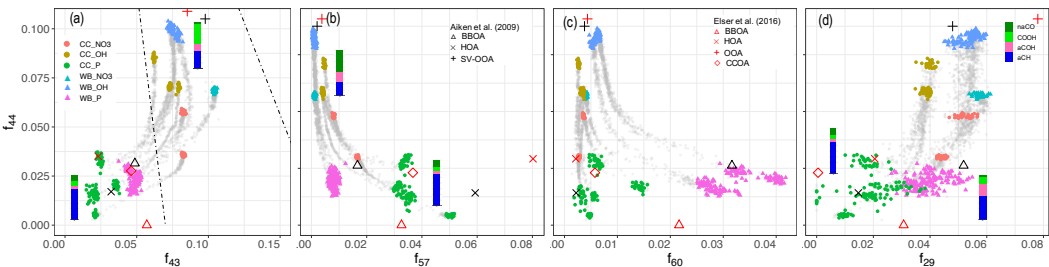

**Figure 5.** Scatter plots of $f_{44}$ against $f_{43}$, $f_{57}$, $f_{60}$, and $f_{29}$. Dashed lines show the outline of the triangle (Ng et al., 2010) and the black shapes show the location PMF factors (Elser et al., 2016; Aiken et al., 2009; Ulbrich et al., 2009). Average FG composition for each category estimated from the MIR spectra are shown beside the category.

## 3.6  Atmospheric biomass burning-influenced aerosols

One of the main purposes of chamber experiments is understanding the characteristics of biomass burning aerosols in the atmosphere. We present a simplified comparison of the MIR spectra of WB aerosols in the chamber with the mean spectra of atmospheric samples affected by burning and other sources. Although the nitrate radical exposures in the chamber experiments were comparable to those of atmospheric samples, no comparable $RONO_2$ bands were observed in ambient samples. This observation implies that organonitrates potentially degrade later in the sequence of reactions due to thermal decomposition (Barnes et al., 1990; Kroll and Seinfeld, 2008) or due to hydrolysis in particle phase (Day et al., 2010; Ng et al., 2017; Liu et al., 2012). As a result, the chamber WB aerosols aged with the nitrate radical were excluded from this comparison due to their very prominent $RONO_2$ bands.

### 3.6.1  Mean spectra

The atmospheric samples were divided into four sub-groups: urban, rural, residential wood burning, and wildfire as described in Sec. 2.4. The individual spectra were baseline-corrected and blank-subtracted, then, were normalized (Euclidean norm) based on their absorbances in 1300–4000 $cm^{-1}$ range. This procedure allowed us to make the spectra with different absorbance magnitudes (i.e. different aerosol mass concentration) comparable. Thereafter, a single mean spectrum was calculated for each sub-group and used as its average representation. These spectra were compared to that of fresh and aged WB OA in the chamber. Since major inorganic compounds of fine aerosols (ammonium, sulfate, and nitrate) are also IR-active and their absorbances overlap with those of organic FGs (especially aCOH), analysis of organic FGs in ambient samples is not always straightforward. To mitigate this problem, ammonium absorbances were subtracted from the mean spectra to obtain the pure contribution of organic compounds.

As can be seen from Fig. 6a, the mean spectrum of rural samples has strong absorptions at ammonium (doublets at 3200 and 3050 $cm^{-1}$), nitrate (1400 $cm^{-1}$), and sulfate (620 $cm^{-1}$) regions, while the peaks attributed to organic compounds are relatively weak (e.g. very weak aCH and carbonyl CO peaks and indistinguishable aCOH absorption due to strong inorganic





absorptions; Fig. 6a). The urban mean spectrum also has strong ammonium and sulfate absorptions, suggesting the abundance
of inorganic compounds in urban sites. The organic signatures in urban spectrum, however, are slightly stronger than that of
rural sites with clear presence of broad carboxylic acid absorption (Fig. 6b). Neither rural, nor urban mean spectra are similar
to that of chamber WB spectra.

In addition to FG identification, we can discuss the presence of specific marker compounds. A weak signature of lignin C=C

The mean spectrum of residential wood burning samples has prominent absorptions of ammonium, sulfate and also nitrate.
Cold weather and high concentration of nitric acid resulting from fossil fuel combustion and biomass burning are believed to
be the primary reasons for the presence of ammonium nitrate on the filters in spite of being relatively volatile (Chow et al.,
2005; Seinfeld and Pandis, 2016). This spectrum has considerably stronger signatures of organic compounds (Fig. 6b). Very
sharp aCH peaks, strong acid COOH absorptions, and a visible aCOH absorption on the left shoulder of the ammonium peak
can be seen in residential wood burning mean spectrum. Comparing this spectrum with that of aged WB in the chamber
reveals their striking similarities. Both spectra have close inorganic-to-organic ratios that result in similar profiles in Fig. 6a. In
addition, both spectra have visible alcohol and acid signatures, which are identified to be important in biomass burning aerosol
composition (Corrigan et al., 2013; Takahama et al., 2011; Russell et al., 2011; Hawkins and Russell, 2010). Nevertheless, the
aCH absorption in the mean spectrum of residential wood burning is stronger than that in the mean spectrum of aged WB in the
chamber. This observation might be attributed to the long-chain hydrocarbons existing in cuticle wax of vegetation detritus that
is absent in these chamber WB experiments (Hawkins and Russell, 2010). Spectral comparison of residential wood burning
aerosols and fresh WB aerosols in the chamber shows that fresh WB lacks inorganics compared to residential wood burning.
Moreover, the relative abundance of aCOH is significantly higher in WB aerosol, indicating the residential wood burning
samples are aged to some degree.

In addition to FG identification, we can discuss the presence of specific marker compounds. A weak signature of lignin C=C
(at 1515 $cm^{-1}$) can be observed in the mean spectrum of residential wood burning (Fig. 6a). In addition, weak levoglucosan
absorbances are visible in some residential spectra. Both mentioned signatures in burning-influenced atmospheric samples are
stronger than that of aged chamber WB and weaker than that of fresh chamber WB aerosols, suggesting that most of these
atmospheric samples have, on average, experienced aging within the range explored by our chamber experiments.

The mean spectrum of wildfire samples is also very similar to that of residential wood burning except having slightly weaker
aCH absorbances. Consequently, the strong COOH absorption and the visible lignin and levoglucosan signatures are also the
characteristics of the wildfire mean spectrum as were for that of residential wood burning. The mean spectra of aged chamber
WB, residential wood burning and also wildfire aerosols (Fig. 6b) are similar to the biomass burning PMF factor obtained by
Hawkins and Russell (2010).

Finally, the samples affected by wildfire are the only atmospheric samples in the IMPROVE network (2011 and 2013) that
have visible (but weak) PAHs signatures, possibly leading to a higher PAH-related toxicity. The areal density of atmospheric
and chamber aerosols collected on PTFE filters is comparable in this work (approximately 10 µg $cm^{-2}$). As a result, the low
absorbance of PAHs in the atmospheric samples implies that most of these compounds are degraded in the atmosphere or
during the filter transportation and storage.





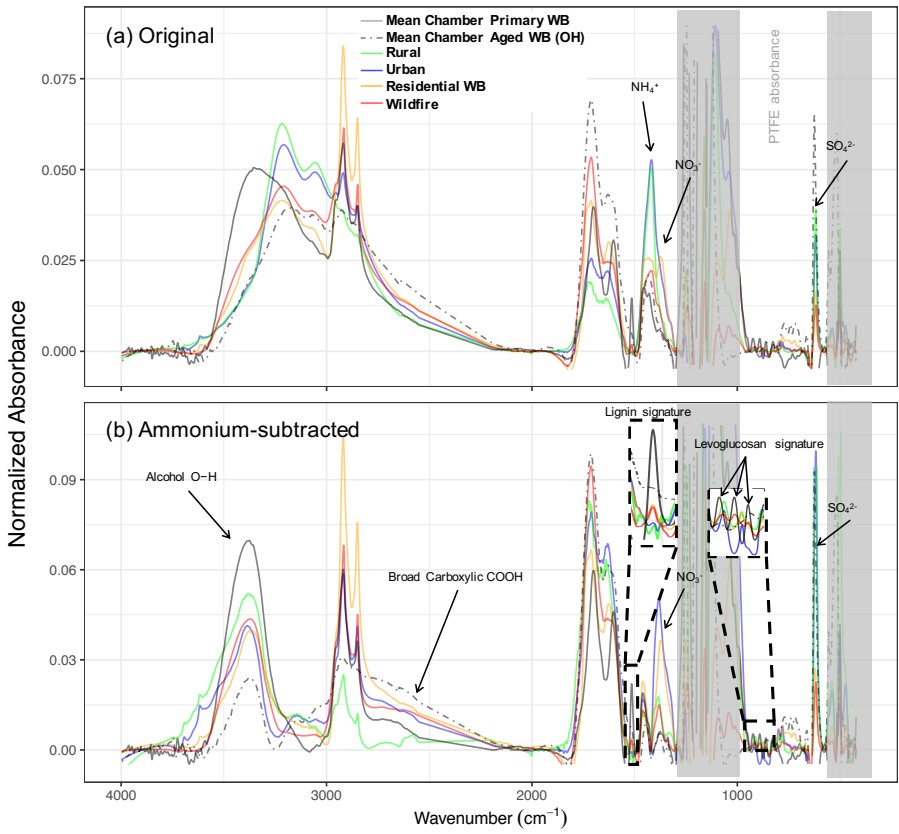

**Figure 6.** Normalized (Euclidean norm) mean spectra of ambient aerosols (rural, urban, residential wood burning, and wildfire) and chamber WB aerosols before (a) and after subtraction of ammonium profile (b). Inorganic nitrate and sulfate absorbances still exist in panel (b).

### 3.6.2 OM:OC ratios

The OM:OC ratio is an important parameter often reported for aerosols collected at monitoring networks (Boris et al., 2019; Ruthenburg et al., 2014; Bürki et al., 2020; Yazdani et al., 2020; Hand et al., 2019; Reggente et al., 2019a). We compared OM:OC ratios from these chamber samples to burning samples identified by Bürki et al. (2020). They used a probabilistic framework on MIR spectra, which resulted OC estimates that were consistent with those from collocated TOR measurements. The average OM:OC estimates for samples influenced by wildfires and residential wood burning were reported to be 1.65 and 1.45, respectively (Bürki et al., 2020) compared to 1.65 and 2 (AMS and MIR spectroscopy obtained close values) for primary and aged WB aerosols in this work. The average OM:OC estimates for these atmospheric burning-influenced samples are clearly closer to that of primary WB aerosols in chamber. Although some biases may exist between the two method due to different estimation methods and calibration standards (Reggente et al., 2019a), this observation is believed to be mainly due to two reasons: First, identified burning-influenced samples in the atmosphere are not as oxidized as the aged WB aerosols in the chamber. Second, sharp aCH peaks, attributed to cuticle wax, are observed in the MIR spectra of ambient fire-influenced





samples (Hawkins and Russell, 2010), while being absent in that of chamber WB aerosols (without bark). The high abundance of aCH group might lower the OM:OC estimate of atmospheric burning-influenced samples.

### 3.6.3  Identification of burning-influenced samples

Biomass burning aerosol composition and its molecular markers continually evolve with aging as discussed in previous sections. This constant evolution poses a substantial challenge to identify the burning-influenced aerosols in the atmosphere and
to quantify the contribution of biomass burning OM to the PM mass. In this section, we assess two different approaches for identification of atmospheric burning-influenced aerosols using MIR spectroscopy.

While each FG is not a marker for any specific source, their proportions could possibly be informative when used with multivariate methods. Cluster analysis by Bürki et al. (2020), for example, identified 45 burning-influenced samples (in 3050 samples) based on their spectral similarity (i.e. FG proportions and organic-to-inorganic ratio), which were supported, to some
extent, by matching their collection time and location with the known burning events (e.g. RIM Fire in California in 2013 and residential wood burning in Phoenix in winter). Nevertheless, the signatures of parent compounds (e.g. lignin) and specific biomass burning markers (e.g. levoglucosan) – direct identifiers of burning – were not considered in their approach. This is because the extended baseline correction and filter subtraction introduced in this work are necessary for identifying these signatures. Moreover, the contribution of these signatures to total variations in the MIR spectra is too minor to be featured for
spectral separation when selecting only a limited number of clusters.

In contrast to FG proportions, tracers are less ambiguous for source identification, but they eventually degrade with aging as discussed in Sec. 3.2. In the tracer approach developed in this work, samples with lignin absorbances above a certain threshold were labeled burning-influenced (absorbances$> 0.0006$ in Euclidean normalized spectra to account for samples with non-negligible contribution of lignin and absorbances$> 0.0004$ in non-normalized spectra to discard samples with low mass
loadings and noisy spectra). Samples with levoglucosan mass (estimated from MIR spectroscopy) contributing more than 5 % to total OM (1.8 of collocated TOR OC estimates) were also labeled burning-influenced (ranging from 5 to 15 % of total OM). We identified in total 173 samples (out of 3050 samples) with the tracer approach, which included the majority of the previously identified burning sample using cluster analysis (38 out of 45) (Fig. 7), suggesting that most of the atmospheric samples with MIR spectrum similar to that of burning aerosols have also visible tracer molecule signatures. There were,
however, 135 samples with levoglucosan and/or lignin signatures above the defined threshold that were not identified by the cluster analysis. The relatively high number of false negatives (i.e. missing burning label when levoglucosan and lignin peaks were present) in the cluster analysis suggests that uncertainties can be high with this approach. There were also 7 samples with negligible levoglucosan and lignin signatures (not identified by the tracer method) that were identified to be burning-influenced by the cluster analysis. The latter discrepancy might exist due to misidentification by the cluster analysis or because
the discussed signatures diminish with extensive aging and cannot be detected by the tracer method. The same could be true for the cluster 10 of Bürki et al. (2020), which was proposed to be possibly influenced by burning, but only 3 out if 28 samples have identifiable levoglucosan or lignin signatures.





The atmospheric burning-influenced samples identified by both methods have a high relative abundance of OM. OM constitutes more than 50 % of fine PM mass for the majority of these sample even at low PM loadings as observed in Fig. 7. However,

a high OM:PM ratio cannot be always a reliable indicator of burning alone since it is not unique to biomass burning and there are many non-burning samples with high OM:PM ratio (Fig. 7).

In summary, the tracer method appears to be able to identify the atmospheric burning-influenced samples including the majority of those identified by the cluster analysis. However, a definitive identification of burning periods is needed to better assess this new approach. Quantifying the contribution of biomass burning OM (fresh and aged) to the PM mass is another

remaining challenge that needs to be addressed in the future work.

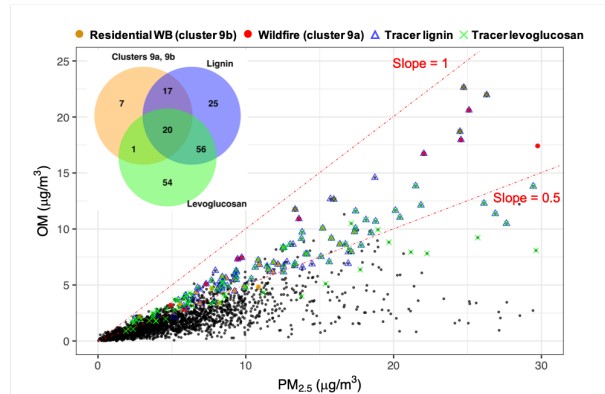

**Figure 7.** Scatter plot comparing total $PM_{2.5}$ and OM mass in atmospheric samples. Orange and red circles indicate residential wood burning wildfire samples identified by cluster analysis (Bürki et al., 2020), respectively. Blue triangles and green crosses shows burning samples identified based on lignin and levoglucosan signatures, respectively. Black circles indicate the existing atmospheric samples in the IMPROVE network (2011 and 2013; approximately 3050 samples). The dotted dashed lines delineate the range of OM mass fraction for samples designated as burning-influenced (the slope of 0.5 is arbitrarily chosen to guide the eye). OM was estimated by multiplying OC by 1.8 (assuming an average OM:OC ratio of 1.8). The Venn diagram at the top left compares the burning samples identified by each method.

## 4   Concluding remarks

In this work, we used MIR spectroscopy and AMS to obtain complementary information about WB and CC aerosols. MIR spectroscopy provides detailed characterization of functional groups and implies that primary aerosols and their parent compounds have similar chemical compositions. This similarity diminishes after aging with appearance of the peaks attributed to

oxygenated species and disappearance of the peaks attributed to the parent compounds. We observed distinct FG compositions for OA based on the emission source (WB and CC), aerosol type (primary and aged), oxidant type, and $NO_x$ concentration. The observed FG composition is informative about the dominant oxidation pathways of WB and CC VOCs and can be used to verify and improve the results of the chemically-resolved SOA formation models.





Dimensionality reduction of the AMS mass spectra reveals similarities between the AMS mass spectra of primary CC OA
and HOA factor, primary WB OA and BBOA factor, and CC and WB OA aged with OH and SV-OOA factor, respectively.
In addition, this analysis suggests that both aerosol source and type are major drivers of variability in the AMS mass spectra. These variations are also consistent with our FG analysis, implying that AMS mass spectra maintain some functional group information in spite of the extensive fragmentation. However, variations due to change in FG composition, occasionally, constitute a small fraction of total variation (stored in higher PCs).

We also found that the MIR spectra of WB aerosols in the environmental chamber are similar to that of ambient samples affected by wildfires and residential wood burning. High abundance organics (especially acids) and existence of peaks attributed to lignin and levoglocosan are the main aspects of this similarity. This result helped us better interpret the MIR spectra of atmospheric samples and was used to aid identification of ambient burning-influenced aerosols.

Finally, it was found that PAHs and aromatics are quantifiable in chamber aerosols using MIR spectroscopy but are not
visible in the spectra of atmospheric samples except for a few burning-influenced instances. Considering the fact that the areal density of aerosols collected on PTFE filters is similar in the atmospheric and chamber samples, this observation implies that most of aromatics and PAHs are degraded in the atmosphere or during the transportation and storage.





*Author contributions.* IEH and ST and AY conceived of the project and manuscript. AB and IEH performed the chamber experiments. AB provided AMS spectra. ND prepared and assembled the filter sampling set-up and took their FT-IR spectra. AMD provided atomized compounds and ambient sample spectra. AY wrote the code for data analysis and post processing, performed the data analysis, prepared laboratory standards and analyzed samples, and wrote the manuscript. ST edited the manuscript and provided regular feedback on the analysis. IEH, ASHP, AB, AMD and ND provided input on the analysis and further editing of the manuscript. ST and IEH provided overall supervision of the project.



*Competing interests.* We declare that no competing interests are present

*Acknowledgements.* The authors acknowledge funding from the Swiss National Science Foundation (200021_172923 and 200021_169787) and the IMPROVE program (National Park Service cooperative agreement P11AC91045).



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
