# Peer review of "Characterization of primary and aged wood burning and coal combustion organic aerosols in environmental chamber and its implications for atmospheric aerosols"

_Atmospheric Chemistry and Physics, 2020_

## Referee Comment (RC1) · Anonymous Referee #1 · 18 Oct 2020

Comment on ACPD script titled as "Characterization of primary and aged wood-burning and coal combustion organic aerosols in an environmental chamber and its implication for atmospheric aerosols" The script states chemical transformations for laboratory-generated wood-burning and coal combustion emissions with respect to OH radical photooxidation and NO3 radical reaction in an environmental chamber. Techniques in combination of in situ AMS measurement and filter-based MIR characterization were applied to extrapolate bulk chemical features of unprocessed and aged samples, the results were also compared with biomass burning related ambient samples to investi-

gate the atmospheric significance and relevance of this study. The dimension reduction/component analysis of the AMS and MIR spectra is very helpful to elucidate bulk functional/structure changes of the aerosol, but the chemical results for the different reaction pathways are of limited novelty and present limited analysis. In addition, the uncertainty caused by the use of a Teflon filter for MIR measurements needs to be discussed. There is no uncertainty analysis or discussion for the AMS and MIR results in the entire manuscript. Nevertheless, this study is an addition to the literature, helping to understand the environmental influence caused by biomass and fossil fuel combustion. Please carefully address the comments before consideration of publication in the journal of Atmospheric Chemistry and Physics. Specific comments: 1. Line 16: the term "chamber WB OA" is confusing, is it unprocessed or aged WB OA? 2. More information concerning the burning conditions is needed, such as combustion efficiency and influence of added kindling. This should be provided in the methods section. Besides, a description of the preparation procedure of the chamber should be added, including particle/gas background, zero-air supply, etc. 3. Line 102: the spectra of the UV lamps should be provided, and the photolysis rate of jNO2 is suggested to compare the chamber environment with ambient conditions. 4. Line 104: Do you mean injecting the H2SO4-NaNO2 solution to the chamber or flushing the evaporated HONO into the chamber? what kind of particle filter? Teflon membrane, quartz fiber, or HEPA filter? 5. In tracing OHexp using butanol-D9, did you consider its wall losses? 6. Line 111: Provide more details on the concentration of the NO3 radical. Is it based on CRDS measurement or box-model simulation? What were the initial concentrations of O3 and NO2 used to achieve the NO3 radical exposure under relatively high humidity (55-60% RH)? Is the NO2:O3 ratio optimized? Was the contribution of NOx emitted by the burning itself considered? What are the estimated wall losses of gaseous species and what are the influences of these wall loss on the results and conclusions? 7. Line 122: AMS collection efficiency should be calibrated using NH4NO3 particles. 8. How the influence of water in the filters on the IR spectra was estimated? Did you condition the filters? 9. Line 219: is there CO2 influence on IR peak in 2400-3400 cm-1? 10. Line

219: change to "carbonyl peak" 11. Line 225: Please explain the reason that a limited number of precursor reactions with NO3 radical lead to the observed less prominent decrease of aCH. There is a place for a more detailed and quantitative analysis of the chemical aging. The normalized results or relative abundances of FG are misleading. For example, aCOH relative abundance in WB aerosol decreases with aging, while the absolute abundances of COOH increase, considering the three-fold increase in OA mass, the absolute concentration of aCOH may also increase. 12. It was stated that phenol, methoxyphenols, and naphthalene are among the most important SOA precursors in WB, can you explain why there were no nitroaromatic products in the MIR for NO3 processed WB. 13. The figures should be improved. Some figures contain too much information and difficult to read. Please pay more attention to color selection in MIR spectra, increase the caption font size, and differentiate overlayers of MIR spectra more clearly. 14. Line 256: is it an absolute or relative abundance of m/z~60 signal that decreased with aging? 15. Line 261: check citation format 16. Explain why the mass spectra include masses only up to m/z~80. Please provide a wider range spectra analysis to show some fragments that are discussed in the manuscript but are not displayed in the figures.

---

## Referee Comment (RC2) · Anonymous Referee #2 · 1 Dec 2020

The manuscript by Yazdani et al. describes the analysis of chamber and ambient biomass burning aerosol. They use two complementary techniques (mid-infrared spectroscopy and aerosol mass spectrometry) to gain insight into the chemical composition of these aerosols as a function of fuel type and aging with hydroxyl and nitrate radicals. This work is a valuable contribution to the literature. After addressing the comments below, I believe this manuscript can be considered for publication in Atmospheric Chemistry and Physics.

General comments

[Figure]

Additional details regarding the chamber setup should be added to the main text and/or the supplement. For example, what wavelength were the UV lights? What impact would these lights have on aerosol aging? What were the concentrations of NO2 and O3 injected into the chamber? What was the source of these gases?

A lot of effort is put into the PCA, but section 3.5 does not provide any overall conclusions. There are conclusions about the relationship between PCA and functional group analysis presented in section 4 that I did not reach from my reading of section 3.5. These conclusions could be presented or discussed more explicitly in section 3.5.

In the discussion of biomass burning tracers, the authors do not mention K+. In proposing a new approach to tracing biomass burning organic matter, context (even if brief) should be provided for the utility of the new approach compared to all other commonly used tracers.

Comparisons between the PM samples collected from the atmosphere and the chamber experiments should be approached with caution. There are several differences between chamber and ambient aerosols that could impact organic aerosol composition, such as different particle size fraction (PM1 vs PM2.5), different fuels, aging conditions, etc. For example, the statement on lines 440-442 about PAHs is highly speculative. Another possible explanation is that conditions in the real environment were not as conducive to PAH formation as were the conditions in the chamber burns.

Specific comments

Line 30: The phrase "ever-increasing importance" is hyperbole. I suggest re-phrasing to "increasing importance"

Line 53: The abbreviation for electron ionization is EI

Line 54: typo in polytetrafluoroethylene. Throughout the manuscript, both "Teflon" and PTFE are used. I suggest eliminating the brand name of Teflon and using polymer names.

**[ACPD](ACPD)**
* * *
Interactive
comment

Figure 3. This figure is difficult to read and interpret. Increasing the size of lines and text would be useful. The multi-colored molecular formulas are confusing—does the extent of each color reflect something about the extent of contribution of each PC? Additional explanation would be useful.

Figure 6. The two colors of grey for the primary and aged chamber aerosol are difficult to distinguish. It also appears that the line for one of them is dashed, though in the legend both lines are solid.

---

## Author Comment (AC1) · 21 Jan 2021

**Response to reviewer comments for manuscript:**
**"Characterization of primary and aged wood burning and coal combustion organic aerosols in environmental chamber and its implications for atmospheric aerosols"**

**Reviewer 1**

Comment on ACPD script titled as "Characterization of primary and aged wood-burning and coal combustion organic aerosols in an environmental chamber and its implication for atmospheric aerosols". The script states chemical transformations for laboratory-generated wood-burning and coal combustion emissions with respect to OH radical photooxidation and $NO_3$ radical reaction in an environmental chamber. Techniques in combination of in situ AMS measurement and filter-based MIR characterization were applied to extrapolate bulk chemical features of unprocessed and aged samples, the results were also compared with biomass burning related ambient samples to investigate the atmospheric significance and relevance of this study. The dimension reduction/component analysis of the AMS and MIR spectra is very helpful to elucidate bulk functional/structure changes of the aerosol, but the chemical results for the different reaction pathways are of limited novelty and present limited analysis. In addition, the uncertainty caused by the use of a Teflon filter for MIR measurements needs to be discussed. There is no uncertainty analysis or discussion for the AMS and MIR results in the entire manuscript. Nevertheless, this study is an addition to the literature, helping to understand the environmental influence caused by biomass and fossil fuel combustion. Please carefully address the comments before consideration of publication in the journal of Atmospheric Chemistry and Physics.

We thank the reviewer for the comments.

**General comments**

1. The dimension reduction/component analysis of the AMS and MIR spectra is very helpful to elucidate bulk functional/structure changes of the aerosol, but the chemical results for the different reaction pathways are of limited novelty and present limited analysis.

   To the best of our knowledge, this is the first time that the functional group composition of primary and aged aerosols is studied using FTIR for major complex combustion emissions in an environmental chamber. We iterate what we believe to be the most novel aspects of this work:

   - We show the extent to which the primary wood burning PM contains signatures of lignin and cellulose pyrolysis products, and the substantial contributions of hydrocarbons to primary coal combustion PM.

   - Main aging pathways for biomass and coal burning are carboxylic acids and non-acid carbonyl formation, respectively.

   - Differences in molecular structure information (FG composition) are observed even after atmospherically-relevant exposures suggests that source or process information is retained in ambient samples and could be used for source apportionment by other instruments (like FTIR), or suite of instruments (involving chromatography and/or soft ionization methods).

- FTIR tracer signatures are used to identify smoke-impacted aerosols in samples from a major US air quality monitoring network.

- From a modeling perspective, the FG analysis of this study indicates that the models of oxidation (models like MCM or GECKO-A) can be evaluated more critically (getting it right for the right reasons) if trajectories beyond O:C can be examined Ruggeri et al. (2016). Furthermore, since the solubility of acid is different from alcohol non-acid carbonyl for instance, distinguishing the pathways of oxygenation has some implications for impacts on health, cloud formation, and eventual fate of the aerosols.

2. There is no uncertainty analysis or discussion for the AMS and MIR results in the entire manuscript.

Uncertainties and limitations of the different instrumentation have been discussed in previous studies and a brief summary is now included in the manuscript:

AMS comparison with semi-continuous thermal optical methods showed that the OC estimates of the two instruments are well correlated (Canagaratna et al., 2007); the $PM_1$ organic and inorganic concentrations have been reported to be reproducible within 25 % (Canagaratna et al., 2007). The collection efficiency for this work has been assumed to be unity based on comparison with size distribution measurements. FTIR calibrations (on PFTE filters) reproduce FG concentrations in laboratory samples within a few percent (Takahama et al., 2013; Ruthenburg et al., 2014) but can diverge more substantially in predictions in more complex ambient samples, depending on the calibration samples and algorithms used (Reggente et al., 2019). For the estimation method used in this work, correlations with collocated thermal optical OC measurements and AMS in field samples have generally agreed within 35% (Gilardoni et al., 2009; Russell et al., 2009; Frossard et al., 2011; Liu et al., 2011; Corrigan et al., 2013; Frossard et al., 2014; Reggente et al., 2019). Debus et al. (2019) have furthermore reported that the measurements made by FTIR instruments are stable over time and instrumental variability are substantially smaller than other uncertainties.

Qualitative and quantitative agreement of FTIR and have been reported in past field campaigns in terms of OA mass, O:C atomic rations (Gilardoni et al., 2009; Russell et al., 2009; Corrigan et al., 2013; Frossard et al., 2014; Faber et al., 2017). FTIR and AMS agreement ($R^2$ 0.92 for OA concentration) in this work is believed to be reasonable regarding the uncertainties of both results and also suggests that variability across the experiments of the same and different categories are real.

We added this sentence to the corrected manuscript:

**The uncertainties of OC and OA mass concentrations derived based on FTIR and AMS have been reported to be within 35 % (Dillner and Takahama, 2015; Gilardoni et al., 2009; Russell et al., 2009; Frossard et al., 2011; Liu et al., 2011; Corrigan et al., 2013; Frossard et al., 2014; Reggente et al., 2019) and 25 %, respectively (Canagaratna et al., 2007). Precision for replicate measurements with the same instrument has been shown to be substantially higher (e.g., Debus et al., 2019).**

Please also note that:

- All results have been reported as ranges across multiple experiments, encompassing the variability across experiments and analytical uncertainties.

- AMS and FTIR results are compared in Figure 1 for OA mass and OM:OC ratio underscores the consistency in the reported findings. This comparison shows that the differences between biomass and coal burning fresh and aged emissions are systematic, much higher than the

variability in burning conditions and instrumental uncertainties.

- We focus on the relative composition of biomass vs. coal burning emissions in this paper, while a detailed comparison between the AMS and FTIR measurements will be presented in a technical note (Yazdani et al., 2020, manuscript in preparation).

**Specific comments**

3. Line 16: the term "chamber WB OA" is confusing, is it unprocessed or aged WB OA?

   The term "chamber WB OA" was changed to "aged chamber WB OA".

4. More information concerning the burning conditions is needed, such as combustion efficiency and influence of added kindling. This should be provided in the methods section. Besides, a description of the preparation procedure of the chamber should be added, including particle/gas background, zero-air supply, etc.

   We have added in the corrected version of the manuscript available information about the combustion conditions and the chamber preparation. The corrected section reads as follows:

   **Four wood burning (WB) and five coal combustion (CC) experiments were conducted in a collapsible Teflon chamber of 6 $m^3$ at the Paul Scherrer Institute (PSI). We studied the effects of fuel source and diurnal/nocturnal aging on the chemical composition of the emissions. The experimental set-up in this work was similar to that used by Bertrand et al. (2017) (Fig. S1).**

   **For the WB experiments, we have followed the procedures developed in Bertrand et al. (2017, 2018a), which favor smoldering-dominated wood fires. Three beech wood logs (approximately 2.5 kg) without bark, and additional 300 g of kindling (beech) were burnt in a modern wood stove (2010 model). The logs were ignited using three fire starters composed of wood shavings, paraffin and natural resins. The moisture content of the logs was measured to be around 11 %. Each burning experiment was started with a lighter followed by immediate closing of the burner door. Emissions past the ignition, in which kindling wood and starters were fully combusted, were injected into the chamber.**

   **In the CC experiments, 300 g of bituminous coal from inner Mongolia (63 % carbon content) was burned. First, the ash drawer of the stove was loaded with kindling wood (Beech), which was ignited and served to ignite the coal. The wood was removed from the drawer after proper ignition of the coal. The emissions past the ignition phase were injected in the chamber via a single injection. Klein et al. (2018) have shown that the temperature in the stove at the starting phase significantly affect the total emission rates of SOA precursors and to a lesser extent their composition. Here, the ignition temperature spanned a similar range as in Klein et al. (2018). Control experiments were performed to evaluate the effect of kindling wood on the emissions where we have followed a similar procedure as for the real experiments but without putting coal in the stove. Resulting emissions after removing the ignited kindling were not different from background for both particle and gas-phases.**

   **After each experiment, the chamber was cleaned by injecting $O_3$ for 1 hour and irradiating with a set of UV lights while flushing with pure air. Then, the chamber was flushed with pure air in the dark for at least 12 hours (similar to the procedure described by Bruns et al., 2015). The pure air injection system consists of a generator (Atlas Copco SF 1 oil-free scroll compressor with 270 L container, Atlas Copco AG, Switzerland) coupled to an air purifier (AADCO 250 series, AADCO**

**Instruments, Inc., USA), which provides a hydrocarbon background of sub 10 ppbC. The background particle- and gas-phase concentrations were measured in the clean chamber before each experiment. Blank experiments were performed, where the chamber was filled with either pure air, or a mix of pure air and ambient air sampled through the heated sampling system and the lights were switched on. In these experiments, approximately 1 $\mu g\, m^{-3}$ of organic aerosol was formed.**

5. Line 102: the spectra of the UV lamps should be provided, and the photolysis rate of $j_{NO_2}$ is suggested to compare the chamber environment with ambient conditions.

   The light spectrum together with $j_{NO_2}$ are provided in (Platt et al., 2013). We provided this information in the corrected version of the manuscript:

   **The OH radical was produced by photolysis of nitrous acid (HONO) continuously injected into the chamber, using UV lights (40 × 100 W, Cleo Performance, Philips). The light spectrum is provided in (Platt et al., 2013); it peaks around 360 nm and has a $j_{NO_2}$ of approximately $10^{-2}$ s$^{-1}$.**

6. Line 104: Do you mean injecting the $H_2SO_4$-$NaNO_2$ solution to the chamber or flushing the evaporated HONO into the chamber? what kind of particle filter? Teflon membrane, quartz fiber, or HEPA filter?

   HONO, generated through the reaction between $H_2SO_4$-$NaNO_2$, was injected into the chamber. A Teflon membrane filter was used. More details are provided in (Platt et al., 2013).

7. In tracing OHexp using butanol-D9, did you consider its wall losses?

   The procedure followed for the determination of OH exposure using butanol-D9 is detailed in Barmet et al. (2012). Based on control experiments (no dilution nor light on), the butanol-D9 concentrations were found to be extremely stable, which indicates no evidence for the loss of the compound onto the chamber walls.

8. Line 111: Provide more details on the concentration of the $NO_3$ radical. Is it based on CRDS measurement or box-model simulation? What were the initial concentrations of $O_3$ and $NO_2$ used to achieve the $NO_3$ radical exposure under relatively high humidity (55–60% RH)? Is the $NO_2$:$O_3$ ratio optimized? Was the contribution of $NO_x$ emitted by the burning itself considered? What are the estimated wall losses of gaseous species and what are the influences of these wall loss on the results and conclusions?

   The $NO_3$ radical concentrations were estimated based on the decay rate of phenol. The $NO_2$:$O_3$ ratio we have used is approximately 1, and their concentrations were approximately 50 ppb. The contribution of $NO_x$ from combustion is less than 10 ppb. Controlled experiments where emissions were injected into the chamber without oxidation showed negligible wall losses of SOA precursors. The effects of vapor wall losses of primary semi-volatile species and of oxidized vapors in our chamber are detailed in (Bertrand et al., 2018a) and in (Jiang et al., 2020), respectively. We have provided these information in the corrected version of the manuscript:

   **The concentration of $NO_3$ was inferred from the reactivity of phenol (m/z 96.058, [$C_6H_7OH$]$H^+$) emitted from wood burning and coal combustion ($k_{NO_3}$ = 3.9 × 10$^{-12}$ cm$^3$ molecules$^{-1}$ s$^{-1}$). We calculated an initial concentration of $NO_3$ of 1–2.5 × 10$^7$ molecules cm$^{-3}$.**

   **The effects of vapor wall losses of primary semi-volatile species and of oxidized vapors in our chamber are detailed in (Bertrand et al., 2018a) and in (Jiang et al., 2020), respectively, and are beyond the scope of this paper.**

9. Line 122: AMS collection efficiency should be calibrated using $NH_4NO_3$ particles.

The AMS ionization efficiency is calibrated using $NH_4NO_3$ particles. The AMS collection efficiency was verified using an SMPS and ranged between 0.7–1.1, and therefore assumed to be approximately 1 for our conditions. We have clarified this in the corrected version of the manuscript:

**The AMS ionization efficiency is calibrated using $NH_4NO_3$ particles. The AMS collection efficiency was verified using an SMPS and ranged between 0.7–1.1, and therefore assumed to be approximately 1 for our conditions.**

10. How the influence of water in the filters on the IR spectra was estimated? Did you condition the filters?

The sample chamber of the FTIR instrument was continuously purged with air treated with a purge gas generator (Puregas GmbH) to reduce water vapor and carbon dioxide interferences. The PTFE filters were left in the sample chamber for three minutes before the scan thus liquid water is expected to evaporate due to the dry purge gas. However, Faber et al. (2017) and Frossard and Russell (2012) showed that water absorption by hydrated salts cannot be reduced efficiently by flushing the sample chamber with a dry gas. The same issue exists for liquid water trapped in closed pockets between layers of aerosol or within individual aerosols (Weis and Ewing, 1999). In this work, conditioning or heating filter samples were avoided to minimize losses of semi-volatile aerosol species. In addition, liquid water absorptions at 3350–3650 $cm^{-1}$ (interfering with the alcohol OH band) and 1550–1700 $cm^{-1}$ were inspected carefully for each individual spectrum and found to be insignificant compared to organic absorptions. The following sentence was added to the manuscript:

**FTIR sample chamber was continuously purged with dry air treated with a purge gas generator (Puregas GmbH) to minimize water vapor and carbon dioxide interferences.**

11. Line 219: is there $CO_2$ influence on IR peak in 2400-3400 $cm^{-1}$?

The $CO_2(g)$ bands appear around 2300 $cm^{-1}$ and are relatively sharp and easily distinguishable from acid OH. In addition, $CO_2(g)$ interference is mainly corrected in the blank subtraction step.

12. Line 219: change to "carbonyl peak"

Misspelling was corrected.

13. Line 225: Please explain the reason that a limited number of precursor reactions with $NO_3$ radical lead to the observed less prominent decrease of aCH.

Different reactivity of $NO_3$ with VOCs compared to OH leads to formation of different SOA species when $NO_3$ is used as oxidant. Based on FTIR and AMS measurements, $NO_3$ SOA appears to contain higher amounts aCH leading to a less prominent decrease in OA aCH. This point has been made clearer in the corrected manuscript. Furthermore, this phenomenon is investigated in more detail in the technical note following this paper (Yazdani et al., 2020, manuscript in preparation) by studying the Van Krevelen plots and high-time-resolution time series of OA functional group composition during aging.

14. There is a place for a more detailed and quantitative analysis of the chemical aging.

A more detailed analysis of aging is presented in a follow-up manuscript using statistical methods (Yazdani et al., 2020, manuscript in preparation). The main implications from the findings of this manuscript were more clearly articulated:

- The results of this manuscript overturns the conventional wisdom of Jimenez et al. (2009) that organic aerosols become similar after aging regardless of their source.

In addition:

- FG composition has been quantitative for FGs with available absorption coefficient. FG group analysis made more complete by mentioning the nitro peaks when aging with $NO_3$.

- A quantitative comparison of levoglucosan degradation is provided in Supplement Sect. S6.

- The main objective of this work is characterization of burning aerosols in terms of their bulk FG composition. A more quantitative treatment of emission factors, combustion efficiency, and OA concentration enhancement has been already done by Bertrand et al. (2017) for similar experiments. Chemical fingerprint of biomass burning aerosols has also been discussed by Bertrand et al. (2018a).

15. The normalized results or relative abundances of FG are misleading. For example, aCOH relative abundance in WB aerosol decreases with aging, while the absolute abundances of COOH increase, considering the three-fold increase in OA mass, the absolute concentration of aCOH may also increase.

The absolute abundances have been shown Fig. 1 of the manuscript with their percentage of contribution to avoid confusion. As stated correctly by the reviewer, the absolute concentration of all FGs increase with aging. This has been investigated in more detail by Yazdani et al. (2020, manuscript in preparation). However, the absolute values are not very useful as the main objective of this study is understanding the compositional change of OA with aging. For examples, relative acid concentration increases with aging for WB aerosol, suggesting WB SOA contains more COOH compared to WB POA, which is dominated by aCOH. In addition the absolute values are not corrected for wall losses.

This sentence was added to the Result and Discussion Section to make the point clearer:

**Relative FG abundances are compared in different OA to understand compositional differences due to aerosol source and age.**

16. It was stated that phenol, methoxyphenols, and naphthalene are among the most important SOA precursors in WB, can you explain why there were no nitroaromatic products in the MIR for $NO_3$ processed WB.

Further investigation of the MIR spectra shows that a peak at 1560 cm$^{-1}$ emerges for the $NO_3$-processed WB and to a lower extent for the $NO_3$-processed CC. This peak can be attributed to nitro compounds (Pavia et al., 2008) and most probably nitroaromatic due to its frequency. However, the low absorbance of this peak compared to that of $RONO_2$ (although its exact absorption coefficient is unknown) and low abundance of aromatics suggest that nitroaromatics are not a major part of OA. We provided this information in the corrected version of the manuscript:

**A relatively small peak at 1560 cm$^{-1}$ is also observed in WB aerosols aged by the nitrate radical, which can be attributed to nitoaromatics (Pavia et al., 2008).**

17. The figures should be improved. Some figures contain too much information and difficult to read. Please pay more attention to color selection in MIR spectra, increase the caption font size, and differentiate overlayers of MIR spectra more clearly.

Figure 1: Functional group color scheme used here is the convention used by the FTIR community.

Figure 2: Font size and line width were increased and the amount of text was reduced to help readability.

Figure 3: In order to make the plot more readable and more easily interpretable, the bars were

made thicker with a different color scheme and the important mass fragments were shown with arrows without color coding.

Figure 4: Text size was increased and the amount of text in the legend was reduced.

Figure 5: Figure configuration was changed and its size was increased and the amount of text in the legend was reduced.

Figure 6: The legend was corrected, the font size was increased, and the color of aged chamber WB was changed to black.

Figure 7: The figure was simplified, font size was increased and the amount of text in the legend was reduced.

18. Line 256: is it an absolute or relative abundance of m/z 60 signal that decreased with aging?

Here, we are referring to the absolute abundance. The relative abundance naturally decreases due to SOA formation and the substantial OA concentration increases with aging. However, we found that the absolute levoglucosan signals from AMS and MIR decrease as well, suggesting decrease in levoglucosan concentration in particle phase due to partitioning or heterogeneous reactions. The importance of levoglucosan vapor wall losses and partitioning for depletion of particle-phase levoglucosan has been highlighted by Bertrand et al. (2018b). The informations was clarified in the corrected manuscript:

**"[...]  both showing a consistent decrease of levoglucosan absolute concentration with aging regardless of the type of oxidant."**

19. Line 261: check citation format 16.

The citation format has been corrected.

20. Explain why the mass spectra include masses only up to m/z 80. Please provide a wider range spectra analysis to show some fragments that are discussed in the manuscript but are not displayed in the figures.

PCA analysis has been performed on the whole available range of the mass spectra (m/z 12 to 202). However, for the sake of simplicity, only fragments up to m/z 80 have been shown in Fig. 3 as the majority of heavier mass fragments have low concentration and thus low PCA loadings. A figure containing PCA loadings for heavier mass fragments have been added to the corrected version of the supplement (Fig. S6) and has been referred to in the main text.

**Reviewer 2**

The manuscript by Yazdani et al. describes the analysis of chamber and ambient biomass burning aerosol. They use two complementary techniques (mid-infrared spectroscopy and aerosol mass spectrometry) to gain insight into the chemical composition of these aerosols as a function of fuel type and aging with hydroxyl and nitrate radicals. This work is a valuable contribution to the literature. After addressing the comments below, I believe this manuscript can be considered for publication in Atmospheric Chemistry and Physics.

We thank the reviewer for the encouraging assessment.

**General comments**

1. Additional details regarding the chamber setup should be added to the main text and/or the supplement. For example, what wavelength were the UV lights? What impact would these lights have on aerosol aging? What were the concentrations of $NO_2$ and $O_3$ injected into the chamber? What was the source of these gases?

Additional information about the chamber set-up was provided in the corrected manuscript:

**Burning conditions: For the WB experiments, we have followed the procedures developed in Bertrand et al. (2017, 2018a), which favor smoldering-dominated wood fires. Three beech wood logs (approximately 2.5 kg) without bark, and additional 300 g of kindling (beech) were burnt in a modern wood stove (2010 model). The logs were ignited using three fire starters composed of wood shavings, paraffin and natural resins. The moisture content of the logs was measured to be around 11 %. Each burning experiment was started with a lighter followed by immediate closing of the burner door. Emissions past the ignition, in which kindling wood and starters were fully combusted, were injected into the chamber.**

**In the CC experiments, 300 g of bituminous coal from inner Mongolia (63 % carbon content) was burned. First, the ash drawer of the stove was loaded with kindling wood (Beech), which was ignited and served to ignite the coal. The wood was removed from the drawer after proper ignition of the coal. The emissions past the ignition phase were injected in the chamber via a single injection. Klein et al. (2018) have shown that the temperature in the stove at the starting phase significantly affect the total emission rates of SOA precursors and to a lesser extent their composition. Here, the ignition temperature spanned a similar range as in Klein et al. (2018). Control experiments were performed to evaluate the effect of kindling wood on the emissions where we have followed a similar procedure as for the real experiments but without putting coal in the stove. Resulting emissions after removing the ignited kindling were not different from background for both particle and gas-phases.**

**Chamber cleaning: After each experiment, the chamber was cleaned by injecting $O_3$ for 1 hour and irradiating with a set of UV lights while flushing with pure air. Then, the chamber was flushed with pure air in the dark for at least 12 hours (similar to the procedure described by Bruns et al., 2015). The pure air injection system consists of a generator (Atlas Copco SF 1 oil-free scroll compressor with 270 L container, Atlas Copco AG, Switzerland) coupled to an air purifier (AADCO 250 series, AADCO Instruments, Inc., USA), which provides a hydrocarbon background of sub 10 ppbC. The background particle- and gas-phase concentrations were measured in the clean chamber before each experiment. Blank experiments were performed, where the chamber was filled with either pure air, or a mix of pure air and ambient air sampled through the heated sampling system and the**

lights were switched on. In these experiments, approximately 1 $\mu g\,m^{-3}$ of organic aerosol was formed.

UV lights: The OH radical was produced by photolysis of nitrous acid (HONO) continuously injected into the chamber, using UV lights ($40 \times 100$ W, Cleo Performance, Philips). The light spectrum is provided in (Platt et al., 2013); it peaks around 360 nm and has a $j_{NO_2}$ of approximately $10^{-2}$ $s^{-1}$.

NO$_3$ generation:

The NO$_2$:O$_3$ ratio we have used is approximately 1, and their concentrations were approximately 50 ppb. The contribution of NO$_x$ from combustion is less than 10 ppb. The concentration of NO$_3$ was inferred from the reactivity of phenol (m/z 96.058, $[C_6H_7OH]H^+$) emitted from wood burning and coal combustion ($k_{NO_3} = 3.9 \times 10^{-12}$ $cm^3\,molecules^{-1}\,s^{-1}$). We calculated an initial concentration of NO$_3$ of 1–2.5 $\times$ $10^7$ molecules $cm^{-3}$.

2. A lot of effort is put into the PCA, but section 3.5 does not provide any overall conclusions. There are conclusions about the relationship between PCA and functional group analysis presented in section 4 that I did not reach from my reading of section 3.5. These conclusions could be presented or discussed more explicitly in section 3.5.

This additional paragraph has been added to the PCA Section summarizing the main findings:

The PCA analysis shows that both aerosol source (WB and CC) and type (primary and aged with different oxidants) are responsible for variability in the AMS mass spectra (similar to the MIR spectra). We also found that the primary WB and CC aerosols have similar mass spectra to the BBOA and HOA factors, respectively and OH-aged OA of both sources are similar to the SV-OOA factor. Furthermore, the spectral variations is consistent with our functional group analysis via FTIR, suggesting that the AMS mass spectra maintain some functional group and source class information even after aging; and in spite of the extensive fragmentation (discussed further by Yazdani et al., 2020, manuscript in preparation). However, even at moderate levels of aging of this work, a part of this information only exists in higher PCs, which explain lower variance in the data (e.g. PC2, explains 12 % of variance across source classes and oxidative aging studied in this work, distinguishes OH-aged CC and NO$_3$-aged WB). These findings are consistent with past reports suggesting that AMS is most sensitive to aging (Jimenez et al., 2009), and underscores the challenges in implying challenges for identifying source classes in highly aged atmospheric OA using AMS.

3. In the discussion of biomass burning tracers, the authors do not mention K$^+$. In proposing a new approach to tracing biomass burning organic matter, context (even if brief) should be provided for the utility of the new approach compared to all other commonly used tracers.

This additional information a Figure in the Supplement has been added:

Potassium is considered a good inorganic tracer of biomass burning (Sullivan et al., 2011). Bürki et al. (2020) showed a higher-than-average elemental K:PM$_{2.5}$ mass ratio for the residential wood burning samples identified in the IMPROVE network (2011 and 2013). However, high elemental K concentrations were not observed in wildfire samples. Sullivan et al. (2011) also reported higher water soluble potassium (K$^+$) and water-soluble organic carbon carbon (WSOC) correlations in winter when residential wood burning is the dominant source of biomass burning. In addition, there are some non-biomass burning sources of potassium more likely to be found in urban areas such as incinerators and fly ash (Sullivan et al., 2011).

**The burning-influenced samples identified by the tracer method in this work have higher concentration of elemental K (from X-ray fluorescence) compared to the majority of other atmospheric samples. However, there are also some samples with very high elemental K concentrations that are not impacted by burning and are most probably affected by mineral dust due to having prominent Si−O−H peak above 3500 $cm^{-1}$ (Bürki et al., 2020).**

4. Comparisons between the PM samples collected from the atmosphere and the chamber experiments should be approached with caution. There are several differences between chamber and ambient aerosols that could impact organic aerosol composition, such as different particle size fraction ($PM_1$ vs $PM_{2.5}$), different fuels, aging conditions, etc. For example, the statement on lines 440–442 about PAHs is highly speculative. Another possible explanation is that conditions in the real environment were not as conducive to PAH formation as were the conditions in the chamber burns.

   The conclusion was made more complete considering the caveats. We believe that different size fraction might not explain this difference as atmospheric samples size range covers that of chamber samples. However, particle size distribution differences can also change gas/particle partitioning, uptake rates, and reaction kinetics. This might be considered another instance of different conditions between chamber and environment.

   The corrected sentence reads as follows:

   **[...] this observation suggests that most of aromatics and PAHs are degraded in the atmosphere or during the transportation and storage, though it is possible that the chamber conditions (e.g., fuel, aging, VOC and OA concentrations) are more conductive to PAH formation.**

   **Specific comments**

5. Line 30: The phrase "ever-increasing importance" is hyperbole. I suggest re-phrasing to "increasing importance".

   The phrase was changed to "increasing importance".

6. Line 53: The abbreviation for electron ionization is EI.

   The abbreviation was corrected.

7. Line 54: typo in polytetrafluoroethylene. Throughout the manuscript, both "Teflon" and PTFE are used. I suggest eliminating the brand name of Teflon and using polymer names.

   The abbreviation was corrected and Teflon was replaced by PTFE in the corrected manuscript.

8. Figure 3. This figure is difficult to read and interpret. Increasing the size of lines and text would be useful. The multi-colored molecular formulas are confusing. Does the extent of each color reflect something about the extent of contribution of each PC? Additional explanation would be useful.

   In order to make the plot more readable and more easily interpretable, the bars were made thicker and the important mass fragments were shown with arrows without color coding.

9. Figure 6. The two colors of grey for the primary and aged chamber aerosol are difficult to distinguish. It also appears that the line for one of them is dashed, though in the legend both lines are solid.

   The legend was corrected, the font size was increased, and the color of aged chamber WB was changed to black.